# Optogenetic induction of mechanical muscle stress identifies myosin regulatory ubiquitin ligase NHL-1 in *C. elegans*

Carl Elias Kutzner [1,2,3], Karen Carolyn Bauer [1,2], Jan-Wilm Lackmann [2], Richard James Acton [4], Anwesha Sarkar [5], Wojciech Pokrzywa [5] & Thorsten Hoppe [1,2,3] ✉

Mechanical stress during muscle contraction is a constant threat to proteome integrity. However, there is a lack of experimental systems to identify critical proteostasis regulators under mechanical stress conditions. Here, we present the transgenic *Caenorhabditis elegans* model OptIMMuS (Optogenetic Induction of Mechanical Muscle Stress) to study changes in the proteostasis network associated with mechanical forces. Repeated blue light exposure of a muscle-expressed *Chlamydomonas rheinhardii* channelrhodopsin-2 variant results in sustained muscle contraction and mechanical stress. Using OptIMMuS, combined with proximity labeling and mass spectrometry, we identify regulators that cooperate with the myosin-directed chaperone UNC-45 in muscle proteostasis. One of these is the TRIM E3 ligase NHL-1, which interacts with UNC-45 and muscle myosin in genetic epistasis and co-immunoprecipitation experiments. We provide evidence that the ubiquitylation activity of NHL-1 regulates myosin levels and functionality under mechanical stress. In the future, OptIMMuS will help to identify muscle-specific proteostasis regulators of therapeutic relevance.

Muscle is composed of sarcomeres, which contain parallel myosin (thick) and actin (thin) filaments that are essential for contraction. Mechanical stress from repeated contractions can damage these proteins, disrupting their structure and function[1]. Ultimately, mechanical stress leads to muscle fatigue, reduced healing capacity, and potentially muscle disease[2–6].

Proteostasis in muscle is mediated by specialized protein folding and degradation mechanisms that are tightly coordinated by regulatory factors[7–9]. The Hsp90 co-chaperone UNC-45 regulates the folding and insertion of myosin motor proteins into the growing sarcomere during development[10–13]. UNC-45 was first shown to be required for myofilament lattice assembly in the nematode *C. elegans* and is conserved in other organisms, including humans[14,15]. In *C.*

*elegans*, UNC-45 localizes to myosin in mature myofilaments[16,17], and temperature-sensitive (ts) missense mutations in UNC-45 result in paralysis or an 'uncoordinated' phenotype at the restrictive temperature, along with disorganized myofilaments, reduced muscle myosin levels, and atrophy[11,18]. In zebrafish, muscle-expressed Unc45b shuttles from sarcomere boundaries to myofilaments after myofiber damage induced by cell membrane wounding, cold shock, or heat shock[19]. In humans, we have recently identified pathogenic variants in the *UNC45B* gene that cause a childhood-onset progressive myopathy[20]. Taken together, these data suggest that UNC-45 supports proteostasis not only during muscle development but also after muscle injury.

The ubiquitin–proteasome system (UPS) plays a key role in the degradation of misfolded and aggregated proteins, and UNC-45 has

[1]Institute for Genetics, University of Cologne, Cologne, Germany. [2]Cologne Excellence Cluster for Cellular Stress Responses in Aging-Associated Diseases (CECAD), University of Cologne, Cologne, Germany. [3]Center for Molecular Medicine Cologne (CMMC), University of Cologne, Cologne, Germany. [4]Human Developmental Biology Initiative (HDBI) at Babraham Institute, Cambridge, United Kingdom. [5]Laboratory of Protein Metabolism, International Institute of Molecular and Cell Biology in Warsaw, Warsaw, Poland. ✉e-mail: thorsten.hoppe@uni-koeln.de

been shown to promote the ubiquitin-dependent degradation of misfolded or non-functional myosin[21,22]. *C. elegans* overexpressing UNC-45 have reduced myosin protein levels and decreased motility, both of which can be partially rescued by inhibiting the proteasome[21]. In addition, *C. elegans* with missense mutations in myosin (*unc-54*) and *unc-45* display increased expression of the proteasomal subunit *rpt-3*, along with myosin misfolding and atrophy[22,23]. UNC-45 protein levels are negatively regulated by a ubiquitylation complex of the E3 ligases CHN-1/CHIP and UFD-2 with the ubiquitin-selective segregase CDC-48/p97[24,25]. Accordingly, *chn-1*, *ufd-2* and *cdc-48* mutations can partially rescue the reduced motility of *unc-45* mutants but not myosin (*unc-54*) mutants in *C. elegans*[25,26]. In mammals, several E3 ligases, such as the F-box proteins Atrogin-1/MAFbx, MUSA1/Fbxo30, and Fbxo21, and the TRIM proteins TRIM63/MuRF1, TRIM54/MuRF3, and TRIM32, are involved in quality control and degradation of myosin and other sarcomeric proteins in various catabolic conditions, such as disuse atrophy and recovery, cancer cachexia, and muscular dystrophy[27–30].

Notably, the roles of UNC-45, myosin-directed E3 ligases, and the UPS in contraction-induced mechanical stress during homoeostasis are incompletely understood. To identify factors and mechanisms that respond to mechanical stress in muscle, we established OptIMMuS (Optogenetic Induction of Mechanical Muscle Stress), a novel optogenetic model to induce sustained contractions in *C. elegans* body wall muscle. The significant anatomical and genetic similarities between *C. elegans* body wall muscle and human skeletal and cardiac muscle facilitate the transferability of findings[31,32]. Here, we use a muscle-expressed *Chlamydomonas rheinhardii* channelrhodopsin-2 (ChR2) variant to repeatedly trigger sustained muscle contractions upon blue light exposure. We combined our transgenic model with an in vivo proximity labelling assay to identify regulators of muscle proteostasis during mechanical stress. Using OptIMMuS, we discovered the E3 ligase NHL-1 as a regulator of UNC-45 and myosin/UNC-54 and provide evidence that the ubiquitylation activity of NHL-1 affects the levels and functionality of misfolded myosin.

## Results

### OptIMMuS triggers sustained muscle contraction in *C. elegans*

To systematically uncover responses to prolonged mechanical stress in muscle, we engineered worms with light-inducible muscle contractions. Specifically, we generated transgenic *myo-3p*::ChR2(C128S;H134R) *C. elegans* that stably express a light-gated ChR2 gain-of-function variant[33] in body wall muscle (Fig. 1a–d). The ChR2(C128S;H134R) variant was selected because it promotes sustained muscle contraction in worms exposed to blue light (450–490 nm) in the presence of the cofactor all-*trans* retinal (ATR)[34] and significantly reduces worm body length and motility for approximately 15 min[33] (Fig. 1a, Supplementary Fig. 1a, Supplementary Movie 1). Worms without ATR supplementation served as a negative control, as the expressed ChR2 protein is non-functional (Fig. 1b)[35]. As additional controls, worms supplemented with or without ATR were kept in the dark. To repeatedly trigger sustained muscle contraction in large-scale experiments, synchronized worm populations were treated with varying numbers of consecutive 5-s light pulses interrupted by 20-min dark recovery phases in a customized incubator equipped with programmable blue LED lights (455 nm, Fig. 1c, d). The sustained contractions resulting from ATR treatment and 10 5-s blue light pulses caused vulval rupture with ejection of eggs and germline tissue in approximately 30–40% of transgenic worms (Fig. 1d, e, Supplementary Fig. 1b). During sample collection and before any downstream analyses, these ruptured animals were removed by sedimentation in M9 buffer, as they did not settle and could be discarded with the supernatant. Furthermore, a single 5-s pulse of blue light reduced the mean worm body length to 70.8% compared to control worms (not supplemented with ATR) (Fig. 1f) and caused partial to complete paralysis (Supplementary Movie 1). Thus, our novel OptIMMuS transgenic

model can be used to trigger sustained muscle contraction in *C. elegans* with an intensity that exceeds voluntary muscle movement.

To examine muscle integrity after sustained muscle contraction, we used phalloidin to stain F-actin-containing sarcomeric thin filaments in body wall muscle cells. Briefly, worms were treated with 10 5-s blue light pulses interrupted by 20-min dark recovery phases, fixed with paraformaldehyde immediately after the last pulse, and compared to control worms without ATR supplementation or without blue light exposure. We observed a 20% decrease in the tip-to-tip length of body wall muscle cells in worms with induced, sustained contractions compared to controls (Fig. 1g, h, Supplementary Fig. 1c). Thus, body wall muscle cells do not regain their full resting length after OptIMMuS-triggered contractions. Notably, body wall muscle cells retained their length in *myo-3p*::ChR2(C128S;H134R) control worms treated with ATR without repeated blue light exposure (Fig. 1h). Exposure to white laboratory light during sample collection can also trigger the functional channel, but this contraction is less intense and does not incur the sustained effects of repeated exposure to blue light pulses. We conclude that the repeated sustained contractions of individual body wall muscle cells and sarcomeres induced by OptIMMuS compromise cell length recovery and overall muscle integrity.

### OptIMMuS-triggered sustained contractions induce multiple stress phenotypes

To further characterize the OptIMMuS-induced mechanical stress responses in muscle, we examined the body wall muscle by transmission electron microscopy (TEM). We observed diffuse dense body-to-actin transition zones in some longitudinal sections of worms treated with ATR and 10 5-s blue light pulses compared to control worms without ATR supplementation but otherwise no apparent disruption of sarcomeric structure (Fig. 2a). The subtle impact of contractions on the muscle sarcomere may be due to the oblique arrangement and dynamic crystalline structure of the sarcomere in *C. elegans*, presenting challenges for exact cross-sectional alignment and detailed imaging of the sarcomere[31,36]. We however observed striking effects of sustained contractions on cytoplasmic organelles, such as autophagosomes, the nucleus, and mitochondria in transverse body wall muscle sections (Fig. 2b). For example, the number and size of mitochondria per body wall muscle cell changed in cross-sections of OptIMMuS-treated worms compared to control worms without ATR supplementation (Supplementary Fig. 2a, b). Using the transgenic reporter *myo-3p*::TOMM-20-mKate, which fluorescently labels mitochondria in body wall muscle, fragmentation of the mitochondrial network was observed in worms treated with ATR and 10 5-s blue light pulses (Fig. 2c). In addition, we observed an increased number of autophagosomes after sustained contractions in TEM consistent with removal of damaged mitochondria by mitophagy[9,37] (Supplementary Fig. 2c, example mitophagosome in Fig. 2b). Taken together, although we cannot exclude that some of these changes reflect prolonged calcium influx through the channelrhodopsin, these data demonstrate multiple cellular stress phenotypes in body wall muscle cells in response to optogenetically triggered repeated sustained contractions.

To monitor whole organism energy expenditure, we measured the motility of a population of worms in the immediate 1-h recovery phase after OptIMMuS treatment. We exposed *myo-3p*::ChR2(C128S;H134R) worms to 10, 13, or 25 5-s blue light pulses interrupted by 20-min dark recovery phases, removed worms with vulval rupture due to OptIMMuS treatment, and performed ARENA WMicrotracker analysis. The number of worms on the corresponding control plates was matched prior to the ARENA analysis. Population motility was dramatically reduced in worms recovering from OptIMMuS treatment compared to control worms without ATR supplementation (Fig. 2d, see also velocity data of individual worms in Supplementary Fig. 1a and repeated light stimulation in Supplementary Movie 2). We propose that this fatigue phenotype observed during

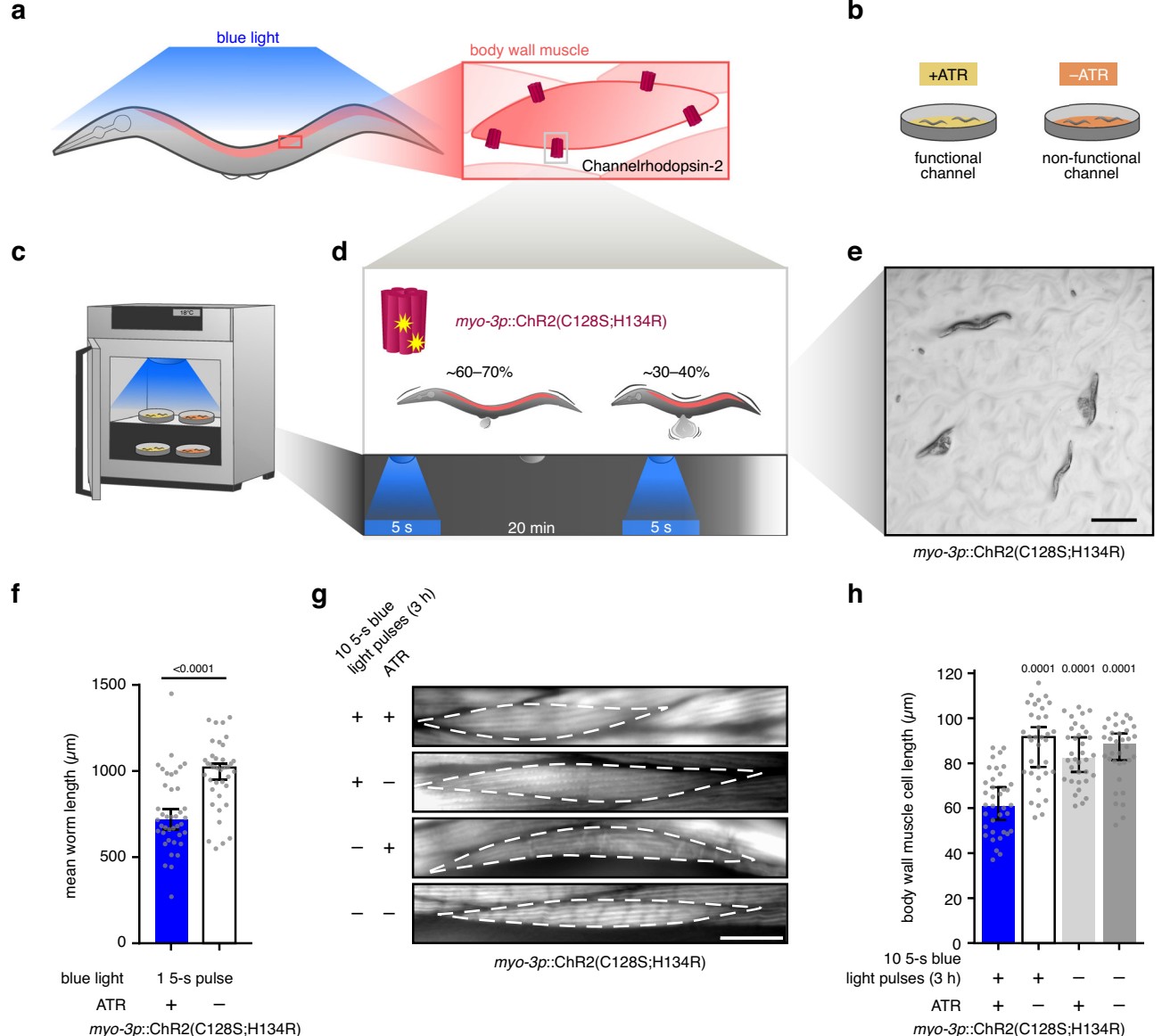

**Fig. 1 | OptIMMuS triggers sustained muscle contraction in *C. elegans*.**
**a** Transgenic worms express *Chlamydomonas rheinhardii* channelrhodopsin-2 (ChR2) in body wall muscle (BWM) cells enabling muscle contraction upon blue light exposure. **b** ChR2 is rendered functional by supplementing the rhodopsin cofactor all-*trans* retinal (ATR), while worms without supplementation of the cofactor serve as control, since the expressed channel remains non-functional. **c** An incubator was equipped with programmable blue LED lights (455 nm) for simultaneous treatment of synchronized worm populations in large-scale experiments at defined time points. As additional controls, worms supplemented with or without ATR are kept in the dark in the same incubator. **d** Transgenic worms expressing the ChR2(C128S;H134R) variant with slow closure kinetics are treated with varying numbers of consecutive 5-s blue light pulses interrupted by 20-min dark recovery phases to repeatedly trigger sustained contractions that paralyse the worms and rupture approximately 30–40% of the individuals. **e** Bright-field light micrograph of

ChR2(C128S;H134R)-expressing worms on ATR after a single 5-s blue light pulse. Scale bar: 500 μm. **f** Bar graphs of mean body lengths of 40 individual worms showing sustained worm body contraction after a single 5-s blue light pulse. Values are median (*n* = 40) with 95% CI; *p*-value compared to control without ATR in two-tailed unpaired Student's t-test. **g** Fluorescence micrographs of rhodamine-phalloidin-stained BWM cells in worms after 10 5-s blue light pulses interrupted by 20-min dark recovery phases. A single BWM cell is outlined with an interrupted white line. Scale bar: 20 μm. **h** Individual BWM cell lengths measured tip-to-tip from images obtained as described in **g** quantifying BWM cell contraction. Values are median (*n* = 36, 35, 34 and 34 cells, left to right, in a total of *n* = 12 animals imaged in *n* = 3 independent experiments) with 95% CI; *p*-values compared to treatment in one-way ANOVA and Dunnett's multiple comparisons test. Source data are provided as Source Data file.

the initial recovery period is a consequence of exhausted energy production in the morphologically abnormal mitochondria observed in the body wall muscle cells of OptIMMuS-treated worms.

To define the organismal stress responses induced by OptIMMuS, we performed whole worm transcriptomics and proteomics experiments. We took advantage of our large-scale setup and collected samples from populations of synchronized worms after repeated, sustained contractions by 25 5-s blue light pulses interrupted by 20-

min dark recovery phases (total treatment time of 8 h) for both omics approaches in parallel (Fig. 2e). We integrated mass spectrometry-identified proteins, referenced against a custom-built *C. elegans*-specific protein database based on the nematode online repository WormBase, with polyA-enriched and sequenced mRNA transcripts and found that protein changes largely mirrored transcript changes. Filtering for significantly altered transcripts (*p*-value ≤ 0.05) identified 1292 transcripts (Fig. 2f, Supplementary Data 1), and filtering for

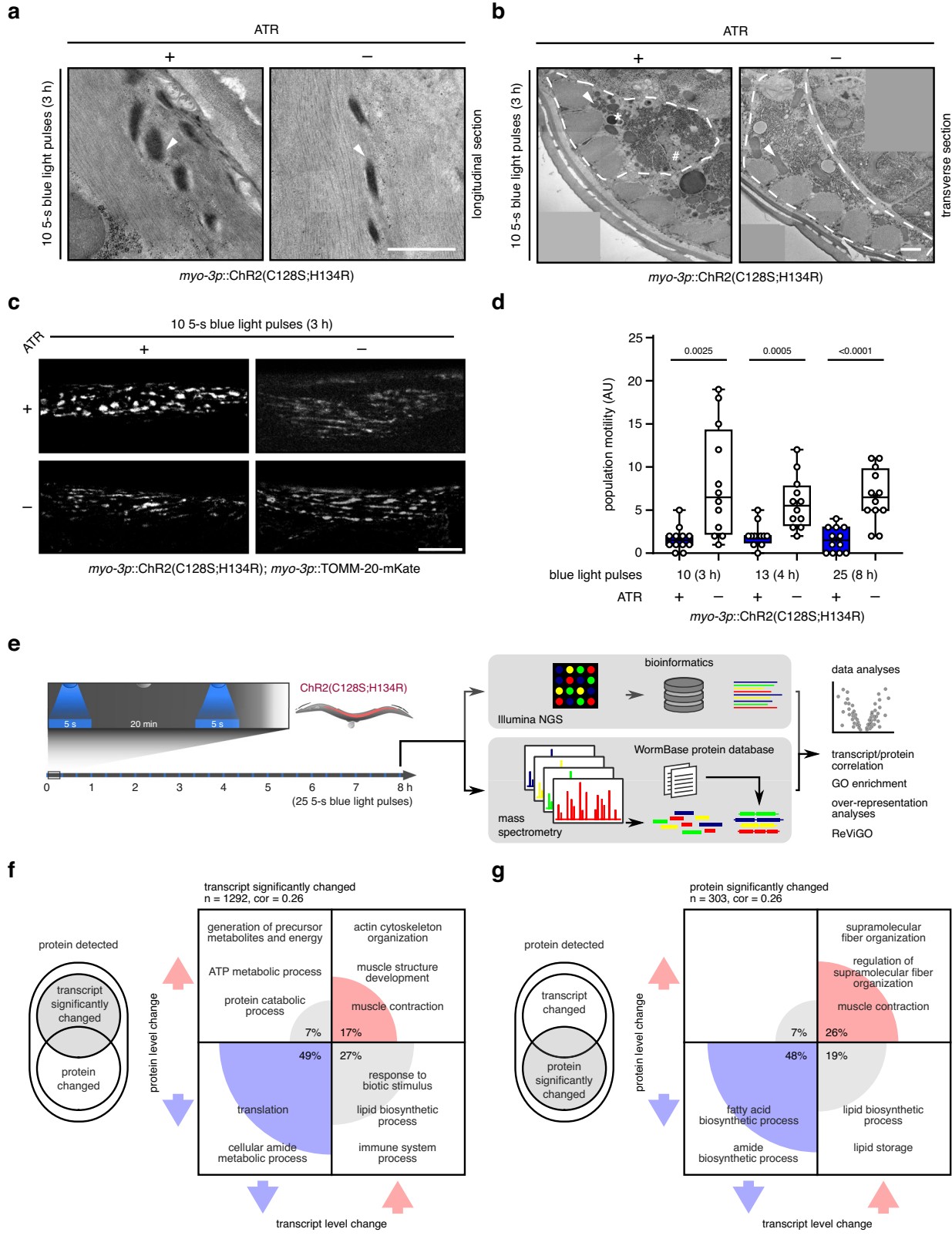

significantly altered proteins (*p*-value ≤ 0.05) identified 303 proteins (Fig. 2g, Supplementary Data 2). For both filtering strategies, we separated parallel and anti-parallel transcript and protein changes into four quadrants and performed over-representation analyses (ORA) for biological process gene ontology (GO) terms on the hits assigned to these quadrants (Supplementary Data 3). Approximately 49% of altered transcripts and 48% of altered proteins were decreased at both

transcript and protein levels and represented biosynthetic processes such as translation, peptide biosynthesis, and fatty acid biosynthesis (Fig. 2f, g), consistent with general transcriptional and translational attenuation during proteotoxic stress[38]. In contrast, approximately 17% of the altered transcripts and 26% of altered proteins were elevated at both the transcript and protein levels and represented cytoskeleton-, muscle fibre-, and contraction-related biological processes, suggesting

**Fig. 2 | OptIMMuS-triggered sustained contractions stress muscle cells, elicit fatigue, and rewire the transcriptome and proteome. a** TEM micrographs of longitudinal sections of body wall muscle (BWM) of a worm treated with ATR and 10 5-s blue light pulses compared to control without ATR. White arrowhead: dense body-thin filament transition zone. Scale bar: 1 μm. **b** TEM micrographs of transverse sections of BWM as in **a**. A single BWM cell is outlined with an interrupted white line. White arrowhead: typical mitochondrion. White asterisk: example mitophagosome. White hash: nucleus. Scale bar: 1 μm. In **a** and **b**, representative images are shown out of $n = 22$ and 19 cells (left to right) imaged in $n = 2$ individual worms per condition. **c** Fluorescence micrographs of TOMM-20::mKate-labelled mitochondria in BWM cells of worms treated as in **a** and control worms without ATR or without blue light exposure. Scale bar: 10 μm. **d** Population motility counts are decreased in the 1-h recovery phase following the indicated OptIMMuS treatments compared to control without ATR. Box plots indicate median (middle line), mean (plus sign), 25th, 75th percentile (box) and minimum to maximum (whiskers) as well as all individual data points. $n = 3$ independent measurements of 30 worms per plate were combined in the analysis; $p$-values compared to control in two-tailed paired Student's t-test. **e** Parallel collection and analysis of the transcriptome and proteome of worms after 25 5-s blue light pulses. **f**, **g** Proportions of transcript/protein pairs regulated in the same (both up [red] or both down [blue]) or opposite (grey) directions in response to OptIMMuS treatment described in **e**. Correlations calculated between transcript and protein log2-fold changes of significantly regulated transcript/protein pairs, where either the transcript (**f**) or the protein (**g**) is significantly regulated ($p$-value ≤ 0.05 from DESeq2 for transcripts and from two-tailed Student's t-test for proteins), regardless of whether the associated protein or transcript is also significantly regulated. The highest enriched Biological Process GO terms identified in over-representation analyses (ORA) are given for each quadrant. Source data are provided as Source Data file; and transcriptomics/proteomics comparison and quadrant ORA source data in Supplementary Data 1, 2, and 3.

that muscle fibre transcripts and proteins are spared from down-regulation. The ORA further revealed an anti-parallel regulation of candidates involved in lipid biosynthesis, response to biotic stimuli, and the immune system, with 27% of altered transcripts and 19% of altered proteins showing significantly increased transcript levels but decreased levels of the corresponding protein. The ORA also revealed 7% of altered transcripts and 7% of altered proteins with decreased transcript levels despite increased levels of their corresponding proteins, representing catabolic processes, proteolysis, and precursor metabolite and energy production (Fig. 2f, g). In conclusion, OptIMMuS-triggered sustained contractions induced proteome and transcriptome changes resembling stress phenotypes, including decreased levels of proteins involved in translation and biosynthesis, and increased levels of muscle proteins and proteins involved in energy metabolism.

### Proximity proteomics identifies UNC-45 interactors in non-stress and OptIMMuS-induced mechanical stress

To study muscle-specific changes in the myofilament landscape in response to repeated, sustained contractions, we performed proximity proteomics of the chaperone UNC-45, which localizes to myofilaments and is involved in myosin refolding upon muscle damage[16,17,19]. First, we sought to identify proteins in UNC-45 proximity under non-stress conditions without sustained contractions. To this end, we used the promiscuous biotin ligase miniTurbo[39] C-terminally fused to UNC-45, expressed in the body wall muscle, which strongly biotinylates proteins in UNC-45 proximity (dark orange halo in Fig. 3a (i)). Since the UNC-45 fusion ligase can diffuse freely in the cytosol, its continuous biotinylation activity can generate background labelling of cytosolic muscle proteins. To subtract this background, we expressed the promiscuous biotin ligase TurboID[39] without a fusion protein in body wall muscle, which labels all cytosolic muscle proteins impartially (orange muscle cell in Fig. 3a(ii)). As a negative control, we expressed UNC-45 without a fusion ligase to account for the background of endogenous biotinylated proteins and unspecific binding to streptavidin (grey muscle cell in Fig. 3a(iii)). Both the fusion biotin ligase UNC-45-miniTurbo and the free biotin ligase TurboID were expressed at similar levels and produced a comparable biotinylation pattern that was enhanced over endogenous background biotinylation (Supplementary Fig. 4d, e).

We performed streptavidin-based pull-down and label-free mass spectrometry quantification of biotinylated proteins in these samples and followed a ratiometric analysis approach. For each identified protein, we compared its intensity in UNC-45 proximity samples (i) with its intensity in muscle protein samples (ii) and in negative control samples (iii) (Fig. 3b). 107 of 1709 proteins were significantly (adjusted $p$-value ≤ 0.05) enriched in UNC-45 proximity compared to the negative control, 33 of 1709 proteins were significantly enriched in UNC-45 proximity compared to the muscle cytosol, and 14 of 1709 proteins

were significantly enriched in UNC-45 proximity compared to both controls (Fig. 3c, Supplementary Data 4). In addition to the myosin head complex (UNC-54, MYO-3, and MLC-3), the small heat shock proteins HSP-16.11, HSP-16.41, HSP-16.48, and HSP-16.2 were significantly enriched in UNC-45 proximity compared to controls. Previously, transcriptional induction of these *hsp-16* heat shock response genes had been observed in *unc-45* knockdown[40] and an increase in HSP-16 family proteins with concomitant conformational rigidity had been observed in the myosin misfolding mutant *unc-54(e1301)* already at the permissive temperature[41]. Thus, our proximity labelling assay now supports a likely physical interaction of UNC-45 and the myosin head complex with small heat shock proteins of the HSP-16 family in *C. elegans* body wall muscle.

To systematically detect changes in the UNC-45 interaction network upon mechanical stress, we combined OptIMMuS with UNC-45 proximity labelling by crossing the ChR2(C128S;H134R)-expressing strain with the UNC-45 fusion ligase strain (Fig. 3d). We subjected these worms to sustained body wall muscle contractions by 13 5-s blue light pulses interrupted by 20-min dark recovery phases with (iv) or without (v) ATR supplementation and repeated the streptavidin-based pull-down with subsequent label-free mass spectrometry quantification of biotinylated proteins (Fig. 3d). We identified 1862 proteins that were biotinylated by the UNC-45 fusion ligase in this experiment (Supplementary Data 4). To again control for the background of endogenous biotinylated proteins and unspecific binding to streptavidin in the new data set, we combined the two UNC-45 proximity proteomics experiments in subsequent analyses (Fig. 3e, Supplementary Fig. 4a, Supplementary Data 4). We detected 1512 biotinylated proteins in both the non-stress and mechanical stress UNC-45 proximity experiments, corresponding to an overlap of 81.2% (Fig. 3e, Venn diagram on the left). To validate the combination of the two experiments, we performed a principal component analysis of both experiments together and observed clustering of the negative control samples (iii) away from the biotin ligase samples (Supplementary Fig. 4b). Furthermore, the UNC-45 proximity samples under non-stress conditions (i) clustered close to the UNC-45 proximity samples of the ChR2(C128S;H134R)-expressing strain in the absence of ATR (v), which represent an equivalent non-stress condition (Supplementary Fig. 4c). Thus, we could apply a similar ratiometric analysis approach to first exclude endogenous biotinylated proteins and unspecific binding to streptavidin and then specifically identify muscle proteins enriched in UNC-45 proximity under mechanical stress.

For each of the 1512 proteins detected in both non-stress and mechanical stress UNC-45 proximity experiments, we first compared its intensity in UNC-45 proximity samples (i) and in muscle protein samples (ii) to its intensity in negative control samples (iii). In order to include muscle-expressed proteins in our analysis that are not yet in UNC-45 proximity under non-stress conditions but might move closer under mechanical stress, we considered 123 proteins that were at least

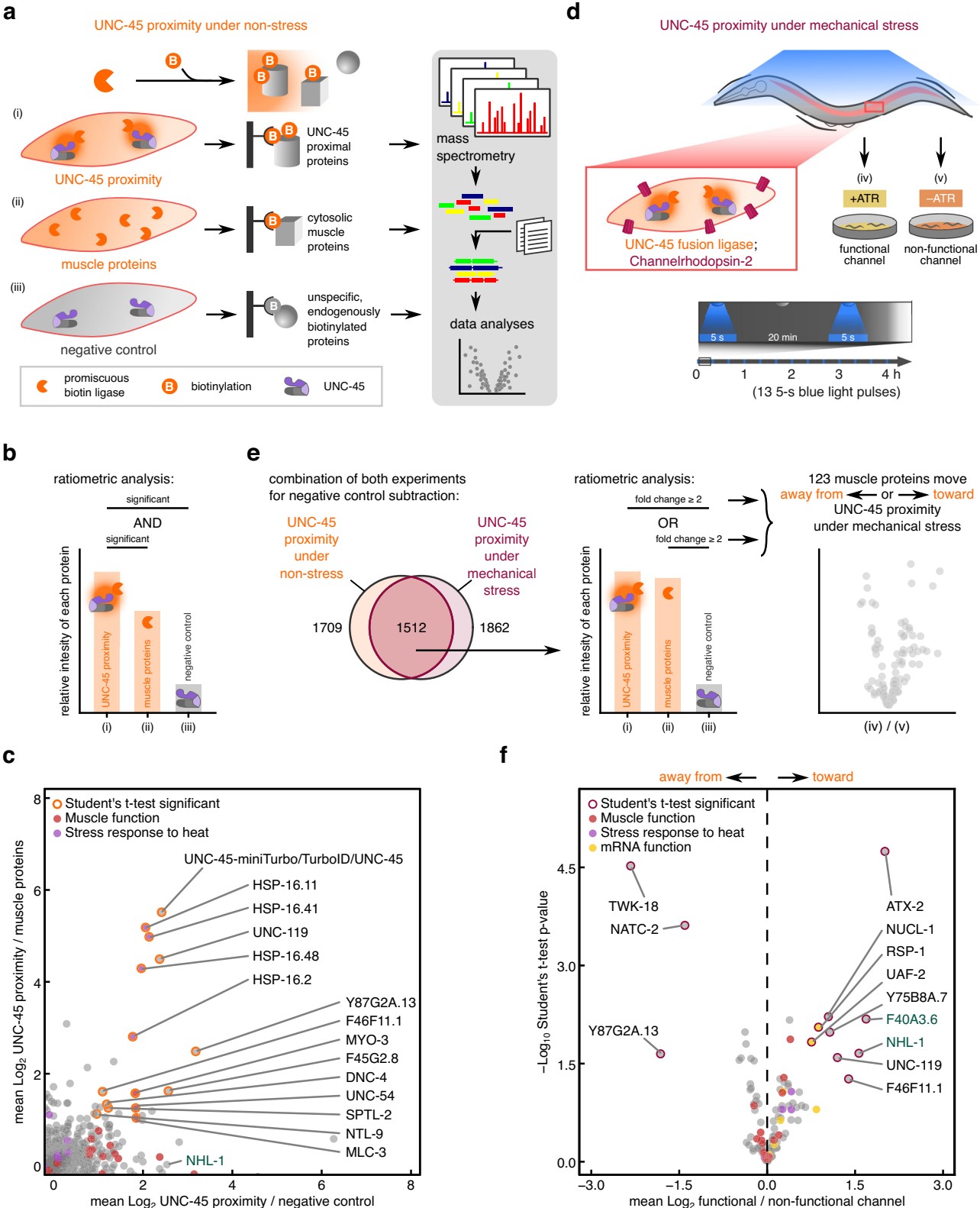

2-fold positively enriched in UNC-45 proximity (i) or among muscle proteins (ii) compared to the negative control (iii) (Fig. 3e, bar graph in the middle). Next, we compared the relative abundance of these proteins in the ChR2(C128S;H134R)-expressing strain with (iv) or without (v) ATR treatment representing movement toward (higher abundance in (iv)) or away from (higher abundance in (v)) UNC-45 proximity under mechanical stress, respectively (Fig. 3e, volcano plot on the right). We found that mechanical stress in muscle shifted the network of UNC-45

proximal proteins toward factors involved in mRNA and proteostasis regulation (Fig. 3f).

## The E3 ubiquitin ligase NHL-1 interacts with UNC-45 and mis-folded muscle myosin

The E3 ubiquitin ligase NHL-1 together with its reported binding partner F40A3.6[42] were identified in UNC-45 proximity under mechanical stress (Fig. 3f), but less so under non-stress conditions

**Fig. 3 | Proximity proteomics identifies UNC-45 interactors in non-stress and OptIMMuS-induced mechanical stress. a** UNC-45 proximity proteomics under non-stress conditions. Proteins biotinylated by the fusion ligase UNC-45-miniTurbo in UNC-45 proximity (i), by the muscle cytosolic TurboID ligase (ii), and endogenously biotinylated or unspecifically binding proteins in a negative control (iii) were pulled down with streptavidin beads and identified by mass spectrometry. **b** Ratiometric analysis and filtering strategy for proteins significantly enriched in UNC-45 proximity versus the negative control and in UNC-45 proximity versus muscle proteins. **c** Scatter plot of identified proteins from **b** shows 14 two-tailed Student's t-test significant proteins in both comparisons marked by an orange outline. Proteins belonging to the WormCat categories "Muscle function" and "Stress response to heat" are marked in red and purple, respectively. **d** UNC-45 proximity proteomics under mechanical stress conditions. Proteins biotinylated by the fusion ligase UNC-45-miniTurbo in UNC-45 proximity in worms subjected to

sustained body wall muscle contractions by 13 5-s blue light pulses with (iv) or without (v) ATR supplementation were pulled down with streptavidin beads and identified by mass spectrometry. **e** Proteins detected in both UNC-45 proximity experiments were combined for negative control subtraction (left). Ratiometric analysis and filtering strategy for proteins at least two-fold positively enriched in both UNC-45 proximity (i) and muscle (ii) versus the negative control (iii) (middle) to compare their movement away from or toward UNC-45 proximity under mechanical stress in a volcano plot (right). **f** Volcano plot of 123 muscle proteins that either moved away from (negative (iv)/(v) ratio) or toward (positive (iv)/(v) ratio) UNC-45 proximity identified 12 candidates significantly changed in two-tailed Student's t-test marked by a red outline. Proteins belonging to the WormCat categories "Muscle function", "Stress response to heat", and "mRNA function" are marked in red, purple, and yellow, respectively. Proximity proteomics source data are provided in Supplementary Data 4.

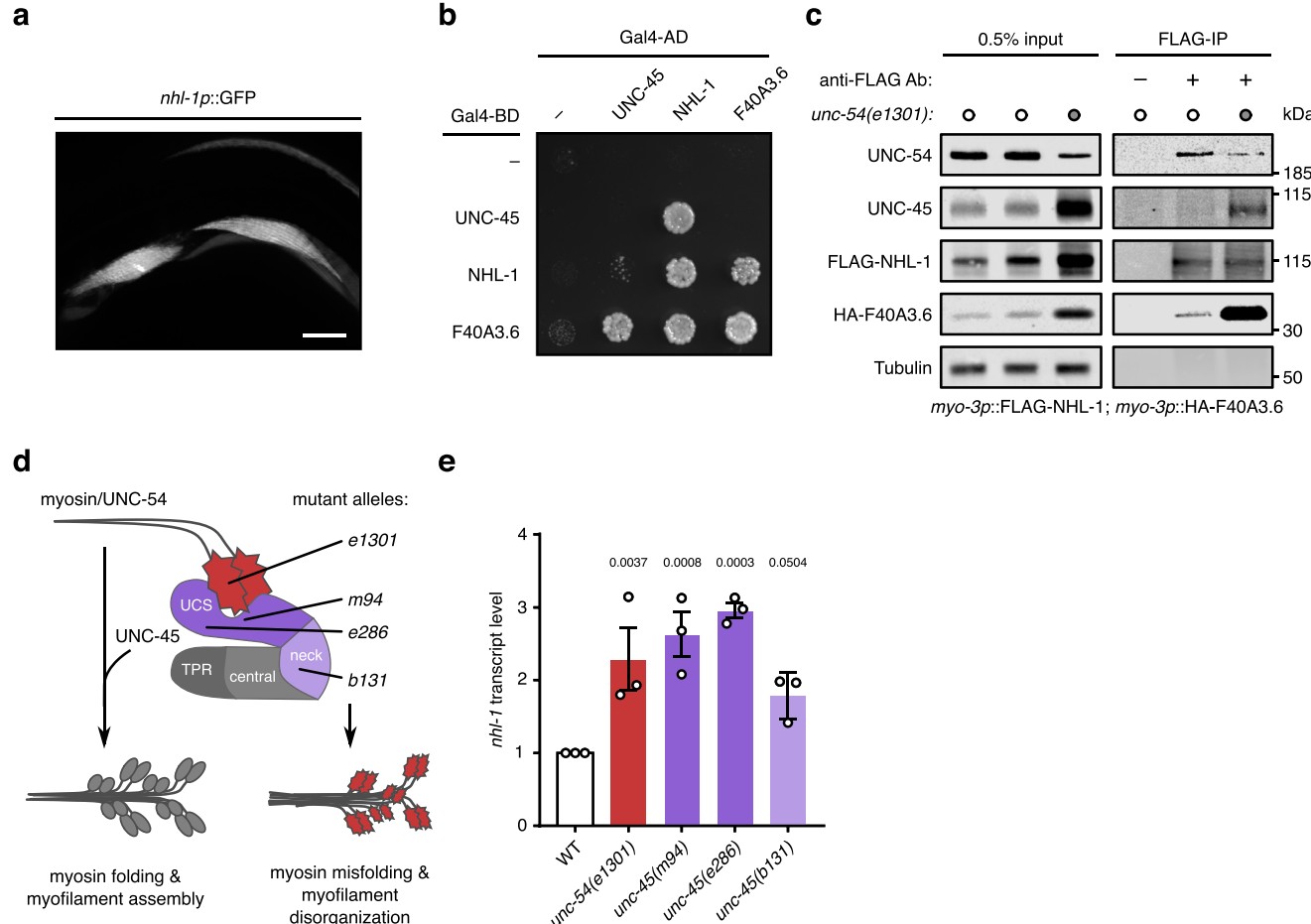

**Fig. 4 | The E3 ligase NHL-1 and its binding partner F40A3.6 interact with UNC-45 and misfolded muscle myosin. a** Fluorescence micrograph of a young adult worm carrying an extrachromosomal *nhl-1p*::GFP transcriptional reporter transgene shows *nhl-1* expression in body wall muscle cells. Representative image of *n* = 2 independent experiments. Scale bar: 20 μm. **b** Yeast grown on an SD/−Leu/−Trp/−His yeast two-hybrid agar plate after 4 days at 30 °C show growth and binding of NHL-1, UNC-45, and F40A3.6. **c** Western blot after immunoprecipitation of FLAG-tagged NHL-1 transgenically expressed in the muscle of wild-type (WT) and

*unc-54(e1301)* worms shows binding of myosin (UNC-54), UNC-45 and transgenically expressed HA-F40A3.6. Representative result of *n* = 3 independent experiments. **d** Myosin folding by the chaperone UNC-45, and location and effect of missense alleles in the *unc-54* and *unc-45* genes. **e** Relative *nhl-1* transcript levels determined by qRT-PCR show increased expression in myosin misfolding mutants compared to WT. Values are mean (*n* = 3 biological replicates) with SEM; *p*-values compared to WT in repeated-measures one-way ANOVA and Dunnett's multiple comparisons test. Source data are provided as Source Data files.

(Fig. 3c, Supplementary Data 4). NHL-1 and F40A3.6 are both reported to be expressed in *C. elegans* body wall muscle[43,44], which we confirmed by proximity proteomics (Fig. 3c, Supplementary Data 4) and, for NHL-1, using the transcriptional reporter *nhl-1p*::GFP (Fig. 4a, Supplementary Fig. 5a). We further confirmed that UNC-45 can bind to NHL-1 and F40A3.6 in a yeast two-hybrid assay (Fig. 4b). To test whether NHL-1 also interacts with the UNC-45 substrate UNC-54 (myosin heavy chain

B, MHC B) in vivo, we generated transgenic strains expressing FLAG-tagged NHL-1 and HA-tagged F40A3.6 in body wall muscle. Indeed, we found that UNC-54/myosin and HA-F40A3.6 both co-immunoprecipitated with FLAG-NHL-1 (Fig. 4c), and that HA-F40A3.6 and FLAG-NHL-1 co-immunoprecipitated with UNC-54/myosin (Supplementary Fig. 5b) from lysates of these transgenic worms. In addition, HA-F40A3.6, FLAG-NHL-1 and UNC-54/myosin all co-

immunoprecipitated with UNC-45 (Supplementary Fig. 5c). These data suggest that UNC-54/myosin, UNC-45, NHL-1 and F40A3.6 can interact in vivo.

To determine whether the interactions between NHL-1, F40A3.6, UNC-54/myosin, and UNC-45 are affected by myosin misfolding, we analysed four conditional loss-of-function mutants that exhibit irreversible myosin misfolding, myofilament disorganization, and decreased motility (Fig. 4d)[11,12,14,41]. In these ts mutants, single missense mutations in the myosin head domain (unc-54(e1301), p.G387R), the myosin-binding UCS domain of UNC-45 (unc-45(m94), p.E781K and unc-45(e286), p.L822F) or the neck domain of UNC-45 (unc-45(b131), p.G427E) lead to myosin misfolding phenotypes that are enhanced by growing the worms at the restrictive temperature of 25 °C. Interestingly, qRT-PCR revealed that endogenous nhl-1 transcript levels were increased in all these mutants but not in the wild-type, at the restrictive temperature (Fig. 4e) indicating a functional relationship between these genes. Similarly, UNC-45, FLAG-NHL-1, and HA-F40A3.6 protein levels were all increased in the unc-54(e1301) background compared to wild-type (Fig. 4c). We also found that higher levels of HA-F40A3.6 co-immunoprecipitated with FLAG-NHL-1 in the unc-54(e1301) background, and observed an increased interaction between UNC-45 and FLAG-NHL-1 in the mutant background (Fig. 4c). Thus, we conclude that misfolding of UNC-54/myosin enhances its association with the E3 ligase NHL-1 and F40A3.6, as well as the interaction between NHL-1 and the myosin-directed chaperone UNC-45.

## NHL-1 regulates the levels and functionality of misfolded myosin

To investigate the roles of NHL-1 and F40A3.6, we used RNAi to knock down nhl-1 and F40A3.6 in the unc-54(e1301) background with misfolded myosin, and identified UNC-54(e1301, p.G387R)-interacting proteins by co-immunoprecipitation mass spectrometry (Supplementary Data 5). The nhl-1 knockdown was more efficient and more pronounced on the UNC-54(G387R) interaction network than F40A3.6 or double knockdown (Supplementary Fig. 6a, b). Gene set enrichment analysis (GSEA) revealed that sarcomeric components were significantly enriched in the UNC-54(G387R) interactors upon nhl-1 knockdown, whereas ribosomal and proteasomal components were reduced (Fig. 5a, b, Supplementary Data 6). Peptides corresponding to the ubiquitin protein (UBQ-1/UBQ-2) were also significantly reduced in the UNC-54(G387R)-interacting pool upon nhl-1 knockdown (Fig. 5a). Thus, these results suggest that, in the absence of NHL-1, misfolded UNC-54(G387R) remains associated with sarcomeric components and is retained in the sarcomere rather than being recycled by the UPS.

To assess the functional consequences of the loss of NHL-1 and F40A3.6, we examined the motility of individual worms, measured as the number of body bends per minute in liquid. Wild-type worms thrashed with approximately 190 body bends per minute and were unaffected by RNAi-mediated depletion of nhl-1 and F40A3.6 at the restrictive temperature of 25 °C (Fig. 5c). Interestingly, expression of FLAG-NHL-1 and HA-F40A3.6 in wild-type muscle reduced the number of body bends to approximately 160 body bends per minute (Fig. 5d). The four myosin misfolding mutants displayed only about 10–20 body bends per minute at the restrictive temperature (Fig. 5c). Depletion of nhl-1 and F40A3.6 significantly increased the number of body bends in the unc-54(e1301) and unc-45(m94) backgrounds, but further decreased the number of body bends in the unc-45(e286) background (Fig. 5c). To date, no apparent difference in the phenotypes and mechanisms of the two close UCS domain mutants m94 and e286 has been reported. Both ts mutations result in myosin misfolding and reduction at the protein level with a compensatory increase in UNC-45 protein levels at the restrictive temperature of 25 °C[18]. In crystal structures, the UNC-45(e286) protein has been reported to have a more rigid myosin-binding UCS domain that is moved closer to the central domain[45]. In the unc-45(b131) background, where UNC-45 binding to NHL-1 is normal but binding to UNC-54/myosin is reduced due to structural

remodelling of the myosin-binding canyon[45] (Supplementary Fig. 6c, d), depletion of nhl-1 and F40A3.6 had no significant effect (Fig. 5c). In addition, UNC-54/myosin protein levels were significantly increased upon depletion of nhl-1 in the unc-54(e1301) background (Fig. 5e, f), but not upon depletion of ufd-2 (Supplementary Fig. 7d, e). UNC-54/myosin protein levels were also significantly increased upon depletion of both nhl-1 and F40A3.6 in the two affected unc-45 mutants, but not in unc-45(b131), while UNC-45 protein levels remained unchanged (Supplementary Fig. 7a, b). Notably, unc-54 transcript levels remained unchanged in all RNAi treatments, excluding transcriptional upregulation of unc-54 (Supplementary Fig. 7c). Taken together, these data suggest that depletion of NHL-1 can promote increased levels of myosin, its retention in sarcomeres, and its functionality, when interacting in a complex with UNC-45 and misfolded UNC-54/myosin. We propose that the depletion of NHL-1 allows UNC-45 additional time to fold the retained myosin, thereby partially increasing its functionality.

## NHL-1 E3 ligase activity is required to regulate myosin levels and function under mechanical stress

To determine the role of the E3 ligase function of NHL-1 in regulating myosin levels and functionality, we used CRISPR-Cas9 technology to mutate the first two cysteine residues in the conserved Really Interesting New Gene (RING) domain of NHL-1 to alanines (nhl-1(syb8175), p.C43A;C46A, Fig. 6a), which effectively abolishes its ubiquitylation activity[46]. Similar to nhl-1 depletion, the RING domain mutation nhl-1(syb8175) increased UNC-54/myosin protein levels in the unc-54(e1301) myosin misfolding mutant and slightly in the unc-45(m94) myosin misfolding mutant worms compared to the single myosin misfolding mutants (Fig. 6b, d). Correspondingly, nhl-1(syb8175) resulted in a significant increase in the number of body bends of the unc-54(e1301) and unc-45(m94) myosin misfolding mutants at the restrictive temperature (Fig. 6c). Thus, NHL-1 E3 ligase activity affects myosin protein levels and functionality under conditions of myosin misfolding stress.

To evaluate the role of NHL-1 E3 ligase activity and the proteasome under mechanical stress in our OptIMMuS model, we crossed the ChR2(C128S;H134R)-expressing strain into the NHL-1 RING domain mutant background nhl-1(syb8175) and subjected the worms to 10 5-s blue light pulses interrupted by 20-min dark recovery phases with or without ATR supplementation and with or without proteasomal inhibition by bortezomib treatment. Notably, the presence of nhl-1(syb8175) or bortezomib significantly reduced the percentage of worms exhibiting vulval rupture after OptIMMuS compared to wild-type (Fig. 6e, h). In addition, the tip-to-tip length of body wall muscle cells was significantly less shortened after OptIMMuS in nhl-1(syb8175) or bortezomib-treated worms compared to wild-type (Fig. 6f, g, i, j), indicating reduced force generation of the induced muscle contractions. In summary, these data suggest that the E3 ligase NHL-1 regulates ubiquitin-dependent myosin turnover via the proteasome and optimal myosin function under mechanical stress (Fig. 7).

## Discussion

In this study, we established OptIMMuS to induce mechanical stress in muscle and used this model to study changes in the proteostasis network associated with mechanical forces and myosin misfolding in C. elegans. OptIMMuS induced disruption of organismal, cellular, and ultrastructural integrity, and altered the whole animal transcriptome and proteome (Figs. 1, 2). A previous study used light-induced contraction by a fast neuronal ChR2 variant to describe the effect of sarcomere component mutants on the mechanics and kinetics of a single muscle contraction and relaxation event[47]. In contrast, our data demonstrate that the OptIMMuS model can be used to analyse not only cellular responses in single cells and tissues but also organismal physiology and phenotypes of synchronized worm populations. C. elegans genetic mutants crossed into the OptIMMuS strain define

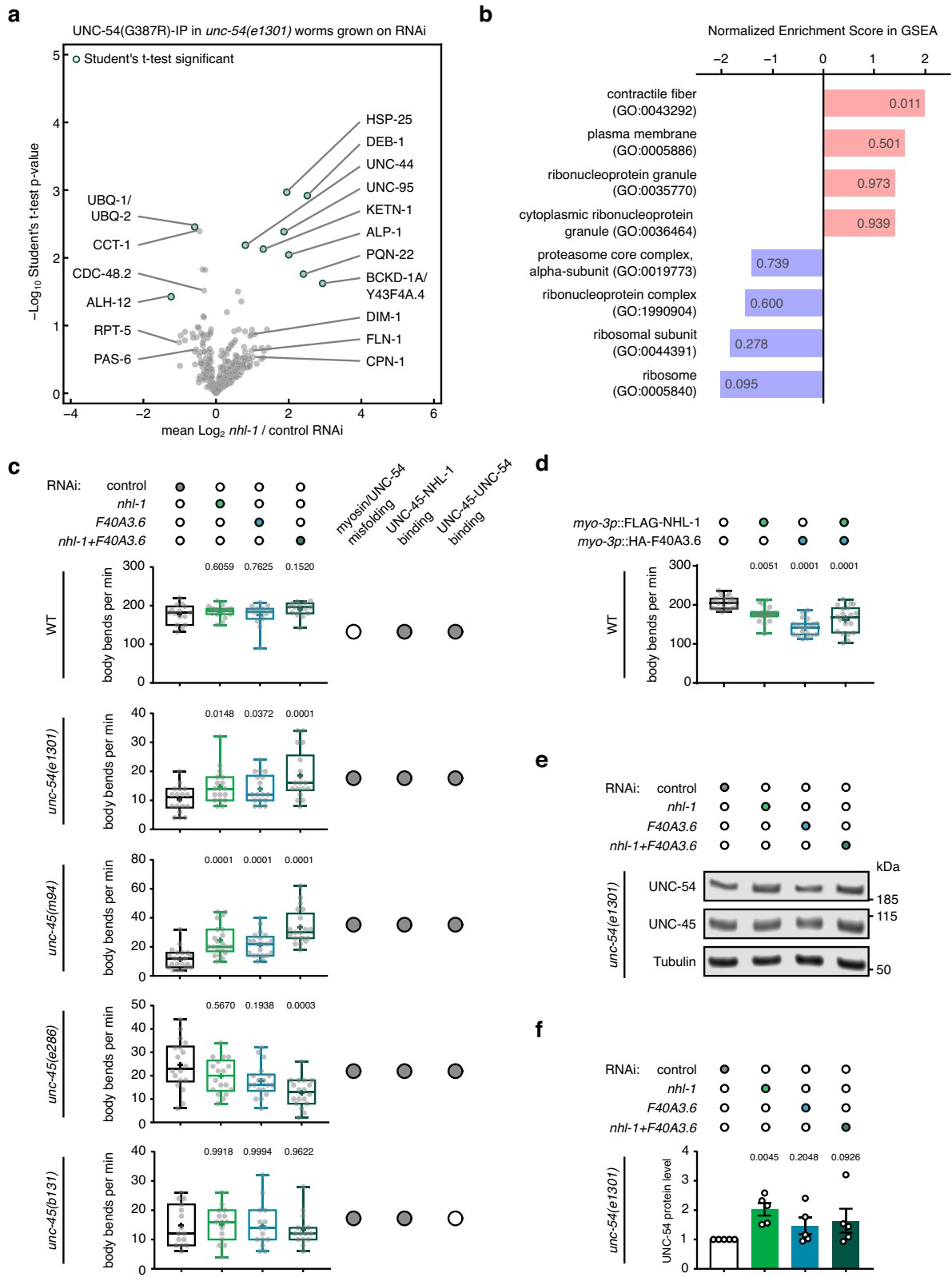

sensitized backgrounds that allow the study of response mechanisms to repeated intense muscle contractions otherwise missed under conventional experimental conditions. When mutations prevent worms from contracting, as in the *unc-45* and *unc-54* mutants, proximity labelling combined with proteomics assays of candidate proteins allows the investigation of dynamic changes in specific interaction networks in the contracting muscle. Using OptIMMuS, we discovered

that the E3 ligase NHL-1 and its binding partner F40A3.6 accumulate in close proximity of UNC-45 and myosin/UNC-54 during mechanical stress, thereby extending the myosin-directed proteostasis network in *C. elegans* (Fig. 3). Our genetic studies showed that the functional and physical interactions are specifically enhanced in global myosin misfolding and muscle atrophy models (Figs. 4, 5), which may explain why NHL-1 has been overlooked under conventional experimental

**Fig. 5 | NHL-1 and F40A3.6 regulate the levels and functionality of misfolded myosin. a** Volcano plot of 332 UNC-54(G387R)-interacting proteins identifies sarcomeric proteins that interact more with myosin, while ubiquitin (UBQ-1/UBQ-2) interacts less with myosin upon *nhl-1* RNAi compared to control. Significantly changed proteins (FDR-adjusted *p*-value ≤ 0.05 in two-tailed Student's t-test) are marked by a green outline. **b** Gene set enrichment analysis (GSEA) of the interactors identified in **a**. GO terms are ranked according to their normalized enrichment score in GSEA and FDR q-values are given in grey. **c** Body bend counts per minute in liquid upon RNAi-mediated depletion of *nhl-1*, *F40A3.6*, or both, compared to control in the indicated genetic background. Body bends of *n* = 15, 18, 21, 18 and 15 individual worms (graphs top to bottom) were counted per condition in *n* = 3, 4, 4, 3 and 3 independent experiments. **d** Body bend counts as in **c** upon transgenic expression of FLAG-NHL-1 and HA-F40A3.6 or both together in the wild-type (WT) background compared to control without transgenes. Body bends of *n* = 20

individual worms were counted per condition in *n* = 3 independent experiments. In **c** and **d**, box plots indicate median (middle line), mean (plus sign), 25th, 75th percentile (box) and minimum to maximum (whiskers) as well as all individual data points; *p*-values compared to control in one-way ANOVA with Dunnett's multiple comparisons test of log2-transformed values. **e** Western blot of UNC-54/myosin, UNC-45, and the housekeeping protein tubulin in lysates from *unc-54(e1301)* myosin misfolding mutants show an increase in myosin/UNC-54 protein levels but not UNC-45 levels upon RNAi-mediated depletion of *nhl-1* compared to control. **f** Quantification of UNC-54/myosin protein levels detected as in **e** normalized to tubulin protein levels and to RNAi control. Values are mean (*n* = 5 biological replicates) with SEM; *p*-values compared to control in repeated-measures one-way ANOVA with Dunnett's multiple comparisons test of log2-transformed values. Source data are provided as Source Data files; and proteomics and GSEA source data in Supplementary Data 5 and 6.

conditions. RNAi-mediated knockdown experiments of *nhl-1* and *F40A3.6* in *unc-45* mutants, each of which uniquely impair UNC-45 chaperone function[12,18,26,45], further suggest that the UNC-45-NHL-1 interaction regulates myosin protein levels and functionality, particularly when myosin is misfolded (Fig. 5). Characterization of a RING domain mutant that abolishes the E3 ligase activity of NHL-1 indicates a ubiquitin-dependent regulation of myosin protein levels and function under conditions of myosin misfolding and mechanical stress (Fig. 6).

Our combined data showing that NHL-1 interacts with misfolded myosin mutants and with UNC-45 during repeated, sustained contractions triggered by OptIMMuS strongly suggest that our mechanical stress model results in local myosin misfolding (Fig. 7). Indeed, several reports demonstrate that myosin undergoes significant conformational changes during cross-bridge cycling and contraction[48–51]. This flexibility renders the myosin head highly susceptible to force-mediated misfolding, as evidenced by its low resistance to heat, which is also generated locally during contraction[52,53]. Furthermore, in vitro atomic force microscopy studies show that myosin unfolding does not follow a defined pathway and that subsequent refolding is impossible without the assistance of the myosin-directed chaperone UNC-45[53,54].

NHL-1 has been reported to have ubiquitylation activity in vitro in cooperation with several E2 enzymes[55]. Despite its strong expression in *C. elegans* muscle (Fig. 4a), *nhl-1* has only been studied in chemosensory neurons in the context of insulin/IGF-1 signalling[56]. It was found to be required for DAF-16/FOXO-dependent organismal protection against heat stress, oxidative stress, and proteotoxicity, but physical NHL-1 interactors have not been identified in vivo. We show that *nhl-1* transcript levels are elevated in myosin misfolding mutants exhibiting myofilament disorganization and atrophy (Fig. 4e), that NHL-1 physically interacts with UNC-54/myosin (Fig. 4c), and that UNC-54/myosin protein levels are increased upon *nhl-1* knockdown or catalytically inactive RING domain mutation in a subset of myosin misfolding mutants (Figs. 5e, f and 6b, d). Similarly, the mammalian myosin E3 ligases TRIM63/MuRF1 and Atrogin-1/MAFbx are upregulated in denervation or disuse atrophy[57–59], bind and ubiquitylate myosin and other sarcomeric proteins[57,60,61], and, when knocked down, prevent myosin loss in conditions where myosin degradation is enhanced[60,62]. The N-terminal peptide sequence of NHL-1 is similar to the myosin-targeting TRIM E3 ligases of the MuRF family (TRIM63 and TRIM54 corresponding to MuRFs 1 and 3). Full-length NHL-1 shares closer sequence and structural similarities with TRIM2, TRIM3, and TRIM32[29,63,64] (Fig. 6a). TRIM2/NARF and TRIM3/BERP are predominantly expressed in the brain, where they both bind to myosin V and ubiquitylate the neurofilament light subunit and the CDK inhibitor p21, respectively, among other substrates[29,65,66]. TRIM32 is expressed in brain and skeletal muscle, binds to myosin, ubiquitylates actin, and mutations in TRIM32 cause limb girdle muscular dystrophy R8 (LGMD2H)[29,30,67,68]. In contrast to MuRF1 and Atrogin-1, TRIM32 is reported to play a minor role during muscle atrophy, but to be

required for skeletal muscle regrowth during recovery from atrophy[69]. Taken together, these findings suggest that NHL-1 is a muscle-specific E3 ligase that targets myosin and possibly other sarcomeric proteins in *C. elegans*.

It has been reported that the *C. elegans* E3 ligase UFD-2 ubiquitylates heat-denatured myosin/UNC-54 presented by UNC-45 in vitro[26]. Interestingly, we did not identify UFD-2 as an interacting partner of UNC-45 or myosin/UNC-54 in our proximity labelling assay under mechanical stress. We have previously reported that the genetic absence of UFD-2 partially rescues the reduced motility of *unc-45(m94)* mutants, whereas the motility of *unc-54(e1301)* mutants is unaffected[25]. Consistently, myosin/UNC-54 protein levels of *unc-54(e1301)* mutants are unaffected by *ufd-2* knockdown (Supplementary Fig. 7d, e), suggesting that only *unc-45* dysfunction can be compensated by *ufd-2* but not the independent misfolding of myosin per se. In contrast, knockdown or RING domain mutation of *nhl-1* partially rescues both motility and myosin/UNC-54 protein levels in both *unc-45(m94)* and *unc-54(e1301)* mutants (Figs. 5 c, e, f and 6b, c, d), suggesting that myosin misfolding is the underlying trigger for NHL-1 to act on myosin protein levels and functionality. We propose that UFD-2 specifically regulates the myosin-directed chaperone UNC-45, whereas NHL-1 directly regulates myosin during mechanical stress, myosin misfolding, and muscle atrophy. Nevertheless, considering the diverse and redundant number of mammalian E3 ligases that regulate myosin, it is likely that there are different ubiquitylation pathways that target myosin for degradation under different physiological conditions, which may also explain why the increase of myosin protein levels by *nhl-1* knockdown or mutation does not fully restore wild-type levels. In contrast to NHL-1, the role of its interactor F40A3.6 in mechanical stress remains unclear, as mammalian orthologs are unknown and depletion of *F40A3.6* had only a marginal effect on myosin protein levels. Interestingly, depletion of *F40A3.6* significantly increased the motility of *unc-54(e1301)* and *unc-45(m94)* mutants, and the body bend data suggest an epistatic interaction with *nhl-1* (Fig. 5c). It could be speculated that upon misfolding, F40A3.6 stops the molecular movement of the myosin motor to allow NHL-1 to extract the misfolded myosin molecule for ubiquitylation.

In conclusion, we present the application of a transgenic *C. elegans* model for characterizing molecular responses to mechanical stress in muscle, which combines the complexity of mammalian in vivo models with the reproducibility and versatility of cell culture experiments. Using OptIMMuS, we identified interactors of the UNC-45 chaperone that regulate myosin-directed proteostasis during mechanical stress, including the E3 ligase NHL-1 and its binding partner F40A3.6. Ultimately, OptIMMuS, in combination with proximity labelling studies of other muscle proteins, will allow the detailed construction of comprehensive proteostasis networks in contracting muscle, thereby enabling the identification of many more as yet unknown regulators of muscle maintenance and the mechanical stress response of therapeutic relevance.

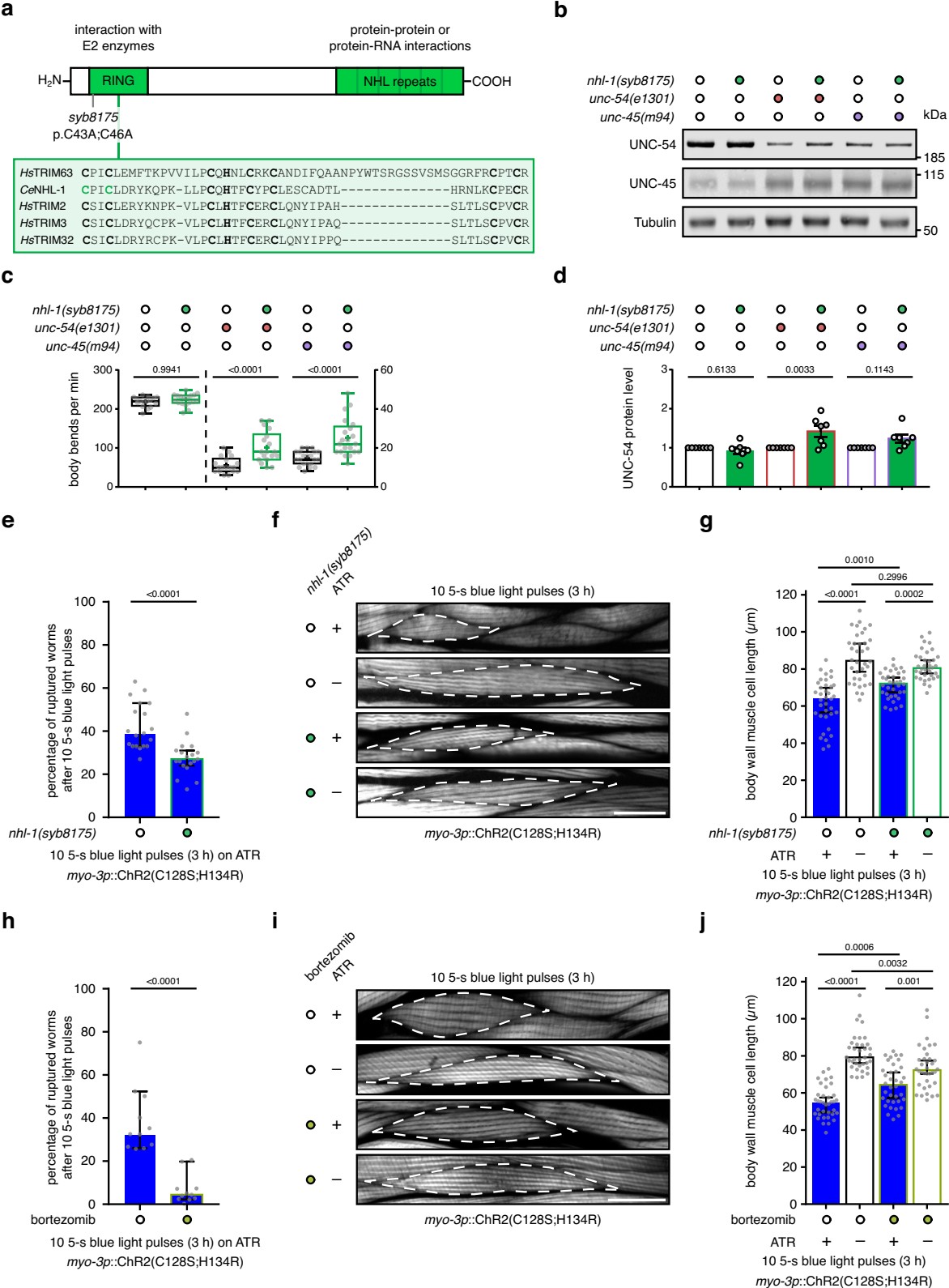

## Methods

### *C. elegans* maintenance

Unless stated otherwise, nematodes were maintained at 15 °C on nematode growth medium (NGM) plates seeded with *Escherichia coli* OP50 as food source according to standard procedures and methods[70]. The N2 Bristol strain (Caenorhabditis Genetics Center) served as wild-type (WT) background. Transgenic strains were generated by microparticle bombardment of plasmids, strains carrying an integrated transgene were recovered, outcrossed several times, and then crossed for experiments. The CRISPR-Cas9 strategy to generate the *nhl-1(syb8175)* (p.C43A;C46A) RING domain mutant strain was developed together with SunyBiotech (Fuzhou, CN) and mutant isolation was performed by the company. Upon arrival, the strain was outcrossed four times into the wild-type background before crossing

**Fig. 6 | NHL-1 E3 ligase activity is required to regulate myosin levels and function under mechanical stress. a** The *nhl-1(syb8175)* strain was generated by exchanging the first two conserved cysteine residues in the N-terminal Really Interesting New Gene (RING) domain of NHL-1 to alanines by CRISPR-Cas9 (p.C43A;C46A) effectively abolishing its E3 ligase activity. **b** Western blot of UNC-54/myosin, UNC-45, and the housekeeping protein tubulin in lysates from the indicated mutant worms. **c,** Body bend counts per minute in liquid of indicated mutant worms. Box plots indicate median (middle line), mean (plus sign), 25th, 75th percentile (box) and minimum to maximum (whiskers) as well as all individual data points. Body bends of $n = 16, 18, 18, 18, 21$ and 21 individual worms (left to right) were counted per condition in $n = 3$ independent experiments; *p*-values compared to control in one-way ANOVA with Sidak's multiple comparisons test of log2-transformed values. **d** Quantification of UNC-54/myosin protein levels detected as in **b** normalized to tubulin protein levels and to control without *nhl-1(syb8175)* presence. Values are mean ($n = 7$ biological replicates) with SEM; *p*-values

compared to control in one-way ANOVA of log2-transformed values with Sidak's multiple comparisons test. **e, h** Frequency of occurrence of percentages of worms with vulval rupture in $n = 19$ (**e**) and 11 (**h**) independent OptIMMuS experiments in the *nhl-1(syb8175)* mutant compared to wild-type (WT) background (**e**) and on bortezomib ($5 \mu M$) compared to DMSO control (**h**). Values are median with 95% CI; *p*-values compared to control in two-tailed Student's t-test. **f, i** Fluorescence micrographs of rhodamine-phalloidin-stained body wall muscle (BWM) cells in WT and *nhl-1(syb8175)* (**f**) or DMSO- and bortezomib-treated worms (**i**) after 10 5-s blue light pulses. A single BWM cell is outlined with an interrupted white line. Scale bars: $20 \mu m$. **g, j** Individual BWM cell lengths measured tip-to-tip from images obtained as described in **f** and **i** quantifying BWM cell contraction. Values are median ($n = 36$ cells in a total of $n = 12$ animals imaged in $n = 3$ independent experiments) with 95% CI; *p*-values compared in one-way ANOVA and Sidak's multiple comparisons test. Source data are provided as Source Data files.

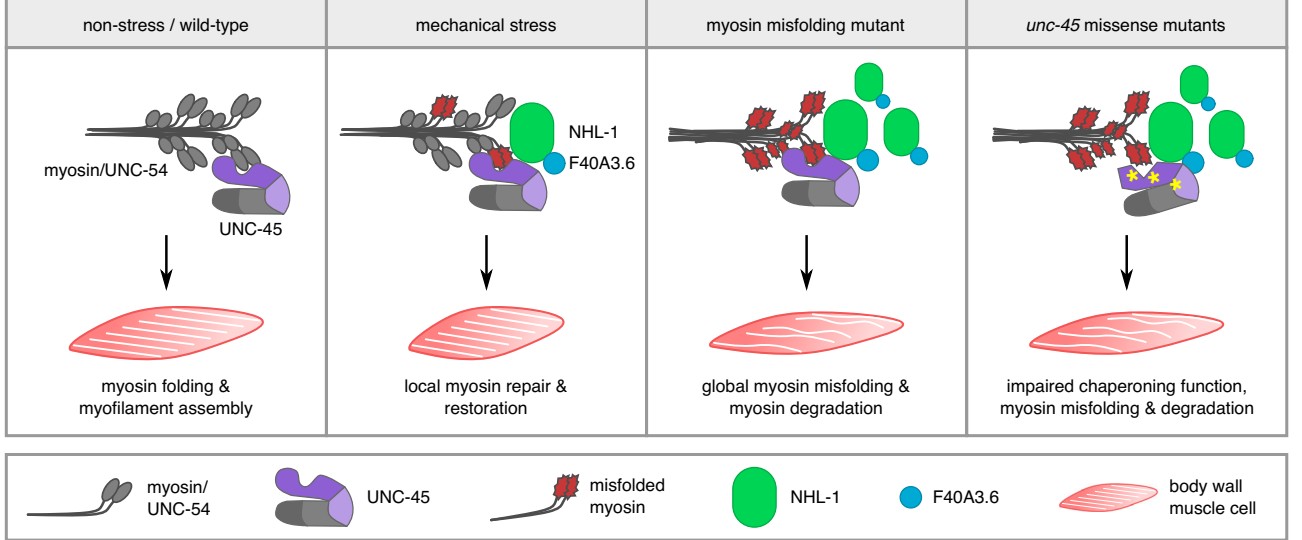

**Fig. 7 | Model for the role of NHL-1 in muscle stress response.** During development and under non-stress conditions, the Hsp90 co-chaperone UNC-45 (purple/grey shape) folds and assembles myosin (grey shapes) into growing myofilaments (parallel white stripes in body wall muscle cells) in muscle. During OptIMMuS-induced mechanical stress, the E3 ubiquitin ligase NHL-1 (green rectangle with rounded corners) and its binding partner F40A3.6 (blue circle) accumulate near UNC-45 in contracting muscle and locally assist in myofilament repair of misfolded

myosin molecules. In conditional myosin loss-of-function mutants, which exhibit global myosin misfolding (red jagged shapes) and myofilament disorganization (wavy white stripes in body wall muscle cells), elevated levels of NHL-1 regulate myosin protein amount and functionality through ubiquitylation. Missense mutations in UNC-45 (yellow asterisks), each uniquely affecting the chaperoning function of UNC-45 and resulting in myosin misfolding, suggest that the interaction of UNC-45, NHL-1, and F40A3.6 is required for myosin regulation.

with myosin misfolding mutants. All worm strains used in this study are listed in Supplementary Data 7. Unless stated otherwise, all experiments were performed on the first day of adulthood and the worms examined were all hermaphrodites. To induce myosin misfolding in ts worms, L1 larvae were grown at 15 °C to L3/L4 larval stage and transferred to 25 °C for 24 h until reaching adulthood prior to body bend measurements and sample collection.

### Cloning and generation of transgenic strains

For the generation of transgenic strains, plasmids containing the *unc-119(+)* selection marker and encoding the respective transgenes *myo-3p*::ChR2(C128S;H134R)-FLAG (OptIMMuS transgenic strain), *unc-54p*::UNC-45-miniTurbo-HA (UNC-45 proximity strain), *unc-54p*::TurboID-HA (muscle proteins strain), *unc-54p*::UNC-45-FLAG (negative control strain), *myo-3p*::FLAG-NHL-1, and *myo-3p*::HA-F40A3.6 were generated using the NEBuilder HiFi DNA Assembly Kit (New England Biolabs) according to the manufacturer's instructions. Fragments were PCR-amplified with Q5 High-Fidelity DNA Polymerase (New England Biolabs) from genomic DNA or from plasmids containing the corresponding open reading frames (ORFs, BioScience)[71]: ChR2 3368 bp plasmid DNA pCS116 (gift from Amelie Bergs and Alexander

Gottschalk), miniTurbo 941 bp plasmid DNA pAS32, TurboID 1178 bp plasmid DNA pAS31, *nhl-1* ORF1 212 bp from ORFeome plasmid DNA, *nhl-1* introns 3–6 999 bp N2 gDNA, *nhl-1* ORF2 2027 bp from ORFeome plasmid DNA, F40A3.6 ORF 774 bp from ORFeome plasmid DNA. Oligonucleotides used in this study are listed in Supplementary Data 8. These constructs were bombarded into *unc-119(ed4)* worms as described previously[72]. Microparticle bombardment was performed using the Bio-Rad Biolistic PDS-1000/HE with ¼" gap distance, 9 mm macrocarrier to screen distance, 28 inches of Hg vacuum and a 1350 psi rupture disc. Approximately 1 mg of 1 µm microcarrier gold beads were coated with 8–10 µg of linearized DNA per bombardment. Animals were allowed to recover for 1 h at room temperature and were then transferred to 90 mm NGM plates seeded with *E. coli* OP50 bacteria. After 3 weeks at 25 °C, motile non-*unc* worms were isolated and screened for homozygosity. All strains that were used in this study are listed in Supplementary Data 7.

### OptIMMuS experiment

For an OptIMMuS experiment, *myo-3p*::ChR2(C128S;H134R)-expressing worms were grown on NGM plates seeded with OP50 supplemented with or without 0.1 mM all-*trans* retinal (ATR) from L1 or latest L4 larval

stage at 18 °C. In parallel, worms grown without cofactor supplementation served as control, since the expressed ChR2 protein remained non-functional. Once the worms were placed on ATR, all plates were kept in the dark. All worms were grown for the last 16 h to adulthood at 18 °C in the OptIMMuS incubator (Sanyo MIR-154), which was equipped in-house with two blue LED lights (Hontiey, each 100 W) spaced 24 cm apart and 30 cm above the sample tray (size: 36 × 54 cm), on which the upturned worm plates are randomly placed for each experiment. The following day, young adult worms were repeatedly illuminated with blue light (455 nm, 10–16 μW per mm$^2$) for 5 s followed by a 20-min dark period. The number of 5-s blue light pulses or total OptIMMuS treatment time is given in the figure legends for each experiment. Blue light illumination was controlled by an FY6800 2-channel DDS arbitrary waveform signal generator (FeelElec) with the following settings: square wave generation, frequency 0.833 mHz, amplitude 7 V, duty cycle 0.417%. The light intensity on varying positions on the sample tray was measured in varying time intervals over the course of the experiments using a ThorLabs Power and Energy Metre PM100D with a S120C Photodiode Power Sensor (400–1100 nm, 50 mW) at the wavelength setting 450 nm (range 1.9 mW) and found to always be in the range of 10–16 μW per mm$^2$, which is above the reported threshold intensity for maximum contraction[33] of 5 μW per mm$^2$. As additional controls in some experiments, worms grown on ATR or without ATR supplementation were kept in the dark in the OptIMMuS incubator for the duration of the experiment. During sample collection after OptIMMuS, ruptured or dead worms were separated by washing the worms in M9 buffer, letting them settle on ice, and discarding the supernatant with floating dead worms, leaving only intact worms in the pellet and in subsequent analyses.

## WormLab measurements

Individual worm behaviour during and after OptIMMuS treatment was observed using the WormLab system (MBF Bioscience). During a 40-min OptIMMuS treatment encompassing two 5-s blue light pulses interrupted by 20-min dark recovery phases, the movement of individual young adult worms was recorded. The frame rate, exposure time, and gain set were kept at 7.5 frames per second. After treatment, the worms were continuously exposed to red light for 80 min, which does not trigger the expressed functional ChR2 protein. Similarly, the movements of the same individual worms were recorded. In the WormLab system, parameters such as track length, worm velocity (centre point speed), total distance travelled by each worm, and average worm body length were analysed every 10 min of the total 2-h measurement. Measurements were repeated in three independent biological replicates. Per replicate, 25–30 worms were recorded and a total of 75 worms from all biological replicates were analysed.

## Phalloidin staining

To measure body wall muscle (BWM) cell length, sarcomeric F-actin was labelled with phalloidin-rhodamine (Invitrogen). Briefly, after 10 5-s blue light pulses interrupted by 20-min dark recovery phases (total OptIM-MuS treatment time 3 h), worms were washed in M9 and fixed in ice-cold 4% (w/v) paraformaldehyde solution for 15 min at room temperature. After permeabilization of the cuticle in a 5% β-mercaptoethanol solution containing 1% Triton X-100, F-actin in BWM sarcomeres was stained with phalloidin-rhodamine (1:200) in 1% BSA (w/v) in PBS containing 0.5% Triton X-100. Stained worms were mounted on glass slides and imaged in the following days using an Axio Imager.Z1 microscope (Zeiss). For each condition, BWM cells were imaged in the same area between the pharynx and vulva, and the tip-to-tip cell length was measured in ImageJ 1.53t. One-way ANOVA with a post-hoc multiple comparisons tests of BWM cell lengths were performed in GraphPad Prism 9.

## Bright-field and fluorescence microscopy

For bright-field images and movies, an Axio Zoom V16 microscope with an Axiocam 506 mono camera (Carl Zeiss Microscopy GmbH) was used, and images were processed using the provided software Zen 2.6 Pro (Carl Zeiss Microscopy GmbH) and ImageJ 1.52a. For close-up fluorescence images of *C. elegans* BWM, the Axio Imager.Z1 (Carl Zeiss Microscopy GmbH) was used with excitation at 550/25 nm (dsRed) or 460–490 nm (GFP) and image detection with Apotome. Worms were washed and fixed in paraformaldehyde as described above for phalloidin staining, mounted directly onto glass slides, and imaged. Images were processed using the provided Zen 2.6 Pro software (Carl Zeiss Microscopy GmbH) or ImageJ 1.53t. To assess mitochondrial morphology in *myo-3*p::TOMM-20-mKate-expressing worms after OptIM-MuS treatment, the confocal laser-scanning microscope ZEISS LSM 980 with Airyscan 2 (Carl Zeiss Microscopy GmbH) with a 594 nm laser was used. Worms were washed and fixed in paraformaldehyde prepared as described above. Obtained images were processed using ImageJ 1.53t.

## Transmission electron microscopy

After 10 5-s blue light pulses interrupted by 20-min dark recovery phases (total OptIMMuS treatment time 3 h), 5–10 *C. elegans* worms were transferred into a membrane carrier (Leica #16707898) prefilled with 20% polyvinylpyrrolidone in PBS. Carriers were frozen using the EmPACT2 (Leica) and samples were stored in liquid nitrogen until further processing. Samples were further processed and analysed by the CECAD Imaging Facility. After freeze substitution using the AFS 2 (Leica) at −90 °C, osmium tetroxide and propylenoxide incubation, and Epon infiltration, individual *C. elegans* worms were removed from the membrane carrier and mounted into flat embedding moulds. The head or tail region was cut longitudinally or transversely into 70 nm ultrathin sections, stained with 1.5% uranyl acetate and 3% Reynolds lead citrate, and imaged on a JEM-2100Plus transmission electron microscope (JEOL). Details can be found in Supplementary Methods. Image analysis was performed in ImageJ 1.52a and statistics in GraphPad Prism 7.

## ARENA population motility measurements

Population motility after OptIMMuS was assessed using the WMicro-Tracker ARENA (NemaMetrix). This instrument measures the motility of small animals (0.5 mm) by recording the number of interruptions of infra-red light microbeams arranged in a large array. After 10, 13, or 25 5-s blue light pulses interrupted by 20-min dark recovery phases (total OptIMMuS treatment time 3 h, 4 h or 8 h), 35 young adult worms were analysed for at least 1 h after treatment at 20 °C instrument temperature. For quantification, the first 15 min of measurement were disregarded as acclimation time of the worms in the instrument. Individual population motility counts, measured every 3.5 min per plate for the first hour after OptIMMuS treatment, were binned in 15 min bins, plotted, and paired Student's t-tests were performed in GraphPad Prism 7.

## Whole worm proteomics and transcriptomics

Worms grown on plates with or without ATR were subjected to 25 5-s blue light pulses interrupted by 20-min dark recovery phases (total OptIMMuS treatment time 8 h) or kept in the dark (four conditions in four biological replicates). Worms were then collected and washed, thereby removing worms with vulval rupture due to OptIMMuS treatment, worm pellets were divided for RNA and protein isolation, snap frozen and stored at −80 °C until further processing. RNA isolation for RNA sequencing was performed on all four independent biological replicates at the same time using Trizol reagent (Thermo Fisher Scientific) and the RNeasy MiniKit (Qiagen) according to the manufacturer's instructions, including DNase I treatment (Qiagen). 2 μg total RNA was sent for RNA-seq. RNA-seq library preparation and sequencing were performed by the Cologne Center for Genomics (CCG), Germany (https://ccg.uni-koeln.de/). Details can be found in Supplementary Methods. Protein isolation for mass spectrometry analysis was performed on all four independent biological replicates together

by sonication on ice in 8 M urea/50 mM TEAB buffer with cOmplete protease inhibitor cocktail (Roche). Protein concentration in cleared lysates was measured using a Pierce BCA Protein Assay Kit (Thermo Fisher Scientific), and 30 μg of total protein was used for Lys-C/trypsin digestion prior to mass spectrometry analysis by the CECAD Proteomics Facility.

## Mass spectrometry analysis

Protein samples for whole worm proteomics, after pull-down of biotinylated proteins, or after UNC-54/ MHC B co-immunoprecipitation were collected in 8 M urea/50 mM TEAB buffer, 30 μg of protein was reduced with 5 mM DTT, alkylated with 40 mM CAA, and digested with 0.5 μg Lys-C and 1 μg trypsin. After overnight incubation, digestion was stopped by adding formic acid to 1 %, peptides were loaded onto equilibrated SDB-RPS stage tips and stored at 4 °C until mass spectrometry analysis. Samples were analysed by the CECAD Proteomics Facility using a Q-Exactive Plus running in DDA coupled to an EASY nLC system running a 60 min gradient (after pull-down of biotinylated proteins and MHC B co-immunoprecipitation) or 150 min (whole proteome) of 0.1% FA against 80% acetonitril with 0.1% FA. The resulting spectra were analysed using MaxQuant 2.0.3[73] with the LFQ option enabled, based on the *C. elegans* WormBase library (PRJNA13758, version WS277, downloaded 05.08.2020)[74]. Results were further processed in Perseus 1.6.15[75], annotated from the Wormbase repository[74] and WormCat[76], and visualized in InstantClue[77]. Details can be found in Supplementary Methods.

## Protein/transcript integration and GO enrichment analyses

All detected proteins were correlated with their corresponding transcripts. Proteomic log2-fold change was calculated from LFQ log2 values and *p*-values for protein differences were taken from the two-tailed Student's t-test calculated by Perseus (blue light plus ATR versus blue light minus ATR). Transcriptomic log2-fold change values and *p*-values were generated for blue light plus ATR versus blue light minus ATR using DESeq2 with an apeglm shrinkage applied to the log2-fold change values. Transcripts and proteins were divided into quadrants based on whether the direction of change was consistent or inconsistent between the transcriptomic and proteomic assays. Candidates were included if there was a significant difference in the transcript or in the protein (two separate analyses). Change type proportion plots were generated as in ref. 78 and show the percentage of transcripts in a given quadrant. Functional enrichment analysis for GO terms of Biological Processes, Cellular Component, and Molecular Function was performed on the candidates in each quadrant using clusterProfiler (https://bioconductor.org/packages/release/bioc/html/clusterProfiler.html)[79,80]. Over-representation analysis (ORA)[81] was performed using the hypergeometric test with a *p*-value cut-off of 0.05. Results were filtered for a q-value less than 0.01 (for transcript significantly changed) or 0.05 (for protein significantly changed), and GO numbers and q-value columns were used as input for REVIGO[82]. The settings for REVIGO were reduction to small (0.5), values represent *p*-value and default settings: remove obsolete GO terms YES (default) and SimRel (default) using the *C. elegans* database (6239). The resulting table for Biological Process GO terms was exported as TSV and the TreeMap as SVG. For the selection of representative GO terms, the five largest sized rectangles (lowest *q*-value) from the larger clusters (with more than one associated GO term) in the TreeMap visualization were selected and three unique GO terms presented in Fig. 2.

## Gene set enrichment analysis

Ranked protein lists generated from the Perseus analysis of UNC-54/ MHC B co-immunoprecipitation and mass spectrometry were subjected to gene set enrichment analysis (GSEA) using GSEA 4.1.0[83]. Default settings for preranked GSEA were maintained, and enrichments were calculated against the *C. elegans* GO and KEGG databases

downloaded from GO2MSIG[84] on July 16, 2021 (http://www.bioinformatics.org/go2msig/). Results were filtered for Cellular Component GO terms and a normalized enrichment score (NES) of > 0.5, and GO numbers and NES columns were used as input for REVIGO[82]. The settings for REVIGO were reduction to small (0.5), higher value is better and default settings: remove obsolete GO terms YES (default) and SimRel (default) using the *C. elegans* database (6239). The resulting table was exported as TSV and the TreeMap as SVG. For the selection of representative GO terms, the five largest sized rectangles (highest or lowest NES) from the larger clusters (with more than one associated GO term) in the TreeMap visualization were selected and four unique GO terms presented in Fig. 5b.

## Yeast two-hybrid

Yeast two-hybrid assays were performed using the Matchmaker GAL4 yeast two-hybrid system 3 (Takara Bio) according to the manufacturer's instructions. Briefly, plasmids expressing fusion proteins of the GAL4 activation domain (AD) or the GAL4 DNA-binding domain (BD) with the proteins of interest UNC-45, NHL-1, or F40A3.6 were generated by cloning the open reading frame of the proteins of interest as a C-terminal in-frame fusion from ORFeome plasmids (BioScience). UNC-45 single point mutant plasmids were generated using a Q5 site-directed mutagenesis kit (New England Biolabs) according to the manufacturer's instructions. All oligonucleotides used in this study are listed in Supplementary Data 8. AH109 yeast cells were transformed with two plasmids, each expressing either an AD fusion protein or a BD fusion protein of interest in varying combinations, using the LiAc/SS carrier DNA/PEG method described previously[85]. To select positive clones, cells were plated on synthetic defined medium (SD; 6.7 g/l yeast nitrogen base, 20 g/l glucose, 0.64 g/l dropout mix (Takara Bio)) depleted of leucine (Leu) and tryptophan (Trp). To identify interactions, selected clones were plated in serial dilutions starting at 0.1 OD at 600 nm on selection plates additionally depleted for histidine (His). Cell growth was monitored at 30 °C for 4–5 days.

## FLAG co-immunoprecipitation

Worms were grown to adulthood on OP50 plates at 15 °C and transferred to 25 °C for 24 h prior to sample collection. Worms were collected and washed, and worm pellets were snap frozen and stored at −80 °C until further processing. Protein was isolated by sonication in NP40 lysis buffer (50 mM Tris pH 7.5, 150 mM NaCl, 1 mM EDTA, 1% NP40, 0.25% sodium deoxycholate) supplemented with cOmplete protease inhibitor cocktail (Roche). Protein concentration in cleared lysates was measured using a Pierce BCA Protein Assay Kit (Thermo Fisher Scientific). 0.5–1.0 mg total protein was diluted in PBS with 0.02% Tween-20 (PBS-T) and incubated with 1 μg mouse anti-FLAG antibody (M2, Sigma) overnight at 4 °C. The next day, magnetic Dynabeads Protein A (Life Technologies) were added and incubated for 1 h at 4 °C. The beads were then washed four times in PBS-T, and bound proteins were eluted by boiling in 2x SDS sample buffer (125 mM Tris pH 6.8, 4% SDS, 4% glycerol, bromphenol blue, 0.05% β-mercaptoethanol) for 10 min at 95 °C and separated by SDS-PAGE. Co-immunoprecipitation experiments were performed similarly using the UNC-54/ MHC B antibody or the UNC-45 antibody. Western blotting was performed using antibodies against FLAG (F7475, Sigma), HA (H6908, Sigma), UNC-45 (not commercially available, Hoppe Lab), UNC-54 (mAb 5-8, DSHB), and tubulin (T6074, Sigma). All antibodies used in this study are also listed in Supplementary Data 9. Proteins were detected by immunoblotting on Immobilon-FL PVDF membranes (Millipore) in an Odyssey scanning system (Li-COR Biosciences).

## RNA isolation and qRT-PCR

RNA isolation from frozen worm pellets was performed using Trizol reagent (Thermo Fisher Scientific) and the RNeasy MiniKit (Qiagen)

according to the manufacturer's instructions, including DNase I treatment (Qiagen). Next, 0.5–2 μg of total RNA was reverse transcribed using the High-Capacity cDNA Reverse Transcription Kit (Applied Biosystems), including RNase Inhibitor (Applied Biosystems) in all reactions. Quantitative real-time PCR (qRT-PCR) was performed using the Luna Universal qPCR Master Mix (New England Biolabs) and the Bio-Rad CFX96 Real-Time PCR Detection System. Two technical replicates were analysed per sample. Gene expression levels were calculated by the $\Delta$Ct method using *cdc-42* and *pmp-3* as housekeeping genes. All oligonucleotides used in this study are listed in Supplementary Data 8.

### RNA interference experiments
To achieve transient *C. elegans* gene knockdown (KD), the RNA interference (RNAi) feeding method was used as described[86]. Briefly, *E. coli* HT115 bacteria expressing the respective double-stranded RNA (dsRNA) of *nhl-1* or *F40A3.6* were taken from the open reading frame (ORF) RNAi feeding libraries[71] provided by BioScience. All RNAi clones were confirmed to contain the correct gene by sequencing. To achieve double knockdown from the same plasmid, the *F40A3.6* sequence was excised from the respective ORFeome plasmid using EcoRV (F40A3.4 ORF, 830 bp) and cloned into the *nhl-1* RNAi plasmid in between the T7 promoters by restriction digestion with AfeI/MscI, removing 90 bp of the *nhl-1* ORF, and retransformed into *E. coli* HT115. As a control, bacteria carrying the empty pPD129.36 vector were used for feeding. For the RNAi treatment, age-synchronized L1 larvae were transferred to RNAi plates containing 100 μg/ml ampicillin and 2 mM IPTG, and seeded with overnight cultures of the respective HT115 clones.

### Body bend motility assay
For body bend assays, individual young adult worms were placed in 1 ml of M9 buffer and body bends were counted over 30 s and converted to one minute.

### Bortezomib treatment
For proteasomal inhibition by bortezomib in an OptIMMuS experiment, *myo-3p*::ChR2(C128S;H134R)-expressing worms were grown on NGM plates supplemented with 10 μM bortezomib or the same volume of DMSO and seeded with OP50 supplemented with or without 0.1 mM all-*trans* retinal (ATR) from L4 larval stage at 18 °C. In parallel, worms grown without ATR supplementation served as control, where the expressed ChR2 protein remained non-functional.

### SDS-PAGE and western blot
Protein samples in SDS sample buffer (125 mM Tris pH 6.8, 4% SDS, 4% glycerol, bromphenol blue, 0.05% β-mercaptoethanol) obtained from co-immunoprecipitation or after RNAi treatment were separated on self-poured Bis-Tris poly-acrylamide gels in MOPS running buffer (50 mM MOPS, 50 mM Tris pH 7.7, 0.1% SDS, 1 mM EDTA) and transferred to Immobilon-FL PVDF membranes (Millipore) using a wet blotting system (GE Healthcare). The transfer was verified using stain-free detection in the ChemiDoc Touch Imaging System (BioRad). Membranes were blocked in 5% (w/v) milk (Roth) in TBS with 1% Tween-20 (TBS-T) and incubated in primary antibody solutions in RotiBlock (Roth) in TBS overnight at 4 °C. The following day, membranes were incubated in fluorescent secondary antibody solutions in RotiBlock/TBS-T supplemented with 0.01% SDS. Proteins were detected with an Odyssey scanning system (Li-COR Biosciences) using the software Image Studio 5.2 (LI-COR Biosciences) and quantified using Image Studio Lite 5.2. Repeated-measures one-way ANOVA and post-hoc multiple comparisons tests were performed on log2-transformed values in GraphPad Prism 7. The antibodies used in this study were as follows (see Supplementary Data 9): mouse monoclonal anti-α-tubulin (B-5-1-2) (Sigma, catalogue number T6074, 1:5,000); mouse monoclonal anti-FLAG (M2) (Sigma, catalogue number F3165, 1:10,000); rabbit anti-HA (Sigma, catalogue number H6908, 1:1000); mouse anti-MHC B (DSHB, catalogue number mAb 5-8, 1:2000); rabbit anti-UNC-45 (Hoppe-lab/Biogenes Berlin, custom antibody, 1:10,000)[22]. Secondary antibodies: IRDye 800CW donkey anti-mouse IgG (H + L) (LI-COR Biosciences, catalogue number 926-32212, 1:10,000); IRDye 800CW donkey anti-rabbit IgG (H + L) (LI-COR Biosciences, catalogue number 926-32213, 1:10,000); IRDye 680 donkey anti-mouse IgG (H + L) (LI-COR Biosciences, catalogue number 962-32222, 1:10,000); IRDye 680 donkey anti-rabbit IgG (H + L) (LI-COR Biosciences, catalogue number 962-32223, 1:10,000).

### Statistics and reproducibility
Statistical details of experiments are described in the legends of the figures in which the relevant results are presented. GraphPad Prism 7 software was used for statistical analyses. In general, statistical tests were performed according to the recommendations of the software and are indicated in the corresponding figure legends. All statistical tests were two-sided. When data points were not normally distributed, data were log2-transformed so that parametric tests could be used. Results are usually presented as median and 95% confidence interval (CI) or mean and standard error of mean (SEM). All representative microscopy images and blots were performed in at least $n = 3$ independent experiments except for the following: TEM was performed as $n = 1$ experiment on $n = 2$ individual worms per condition. Representative images in Fig. 2a and b are shown out of $n = 22$ and 19 body wall muscle cells (left to right). Expression of *nhl-1* in body wall muscle was initially observed visually by fluorescence microscopy. Fixation and imaging was performed in $n = 2$ independent experiments and a representative image is shown in Fig. 4a. No statistical method was used to predetermine sample size. No data were excluded from the analyses. The experiments were not randomized and the investigators were not blinded to allocation during experiments and outcome assessment.

### Reporting summary
Further information on research design is available in the Nature Portfolio Reporting Summary linked to this article.

## Data availability
RNA-seq data that support the findings of this study have been deposited in the Gene Expression Omnibus (GEO) repository under accession code GSE241906. Mass spectrometry data that support the findings of this study have been deposited in ProteomeXchange under accession codes PXD045078 (whole worm proteomics), PXD045081 (proximity proteomics), and PXD045076 (UNC-54/myosin Co-IP). Plasmids and *C. elegans* strains generated in this study will be distributed to other researchers upon request. All data supporting the findings of this study are available within the article, the Supplementary Information or the Source Data and extra data are available from the corresponding author upon request. Source data are provided with this paper.

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

## Acknowledgements

We thank the Caenorhabditis Genetics Center funded by the NIH Office of Research Infrastructure Programs (P40 OD010440), Addgene, Geneservice Ltd, and the Developmental Studies Hybridoma Bank for worm strains, cDNA, plasmids, and antibodies; Amelie Bergs and Alexander Gottschalk for ChR2-expressing worm strains and plasmids; Christopher Kier and Leo Leson for help with setting up the OptIMMuS incubator; Gabriele Stellbrink for technical assistance. We thank the CECAD proteomics facility for assistance with mass spectrometry analyses and support; the CECAD imaging facility, Christian Jüngst, and Felix Gaedke for assistance with transmission electron microscopy and confocal microscopy analyses; and the CECAD bioinformatics facility and Debasish Mukherjee for assistance with transcriptomics data evaluation and analyses. We thank the Cologne Center for Genomics for library preparation and sequencing. We thank Angela Andersen for valuable comments on the final manuscript. We are especially grateful to Kavya Leo Vakkayil for her assistance with the generation of transgenic strains and to all members of the Hoppe Lab, the FOR 2743, and Christian Sommereisen for valuable discussions of our results and the manuscript. This work was funded by the National Science Centre, Poland (grant SONATA BIS number 2021/42/E/NZ1/00190) to W.P. and by the

Deutsche Forschungsgemeinschaft (DFG) in the framework of the German Excellence Strategy – EXC 2030 – 390661388 and FOR 2743; and by the European Research Council (ERC-CoG-616499) to T.H. Diese Arbeit wurde gefördert durch die Deutsche Forschungsgemeinschaft (DFG) im Rahmen der Exzellenzstrategie – EXC 2030 – 390661388, FOR 2743; und durch den European Research Council (ERC-CoG-616499) an T.H.

## Author contributions

C.E.K. designed, performed, and analysed the experiments together with K.C.B. K.C.B. performed and analysed fluorescence and confocal microscopy experiments. C.E.K. and J.W.L. designed, performed and analysed the mass spectrometry experiments with help from K.C.B. R.J.A. performed transcriptomics analysis and transcriptomics-proteomics data integration. A.S. performed and analysed WormLab experiments. W.P. performed and analysed UNC-45 co-immunoprecipitation in *unc-45* mutants. T.H. supervised the experimental design and data interpretation. C.E.K., K.C.B., and T.H. wrote and edited the manuscript. All authors (C.E.K., K.C.B., J.W.L., R.J.A., A.S., W.P., and T.H.) discussed the results and commented on the manuscript.

## Funding

## Competing interests

The authors declare no competing interests.
