## [Peer Review File · Nature Communications]

Optogenetic induction of mechanical muscle stress identifies myosin regulatory ubiquitin ligase NHL-1 in *C. elegans*REVIEWER COMMENTS

Reviewer #1 (Remarks to the Author):

Kutzner et al use optogenetic stimulation with muscle targeted channelrhodopsin to evoke mechanical stress in *C. elegans* muscle. They document the consequences of this stimulation on muscle structure using light and electron microscopy, and report in particular altered number and size of mitochondria, and an increase in autophagosomes. They combine optogenetic muscle stimulation with transcriptomics and proteomics of muscle, to identify mRNAs and proteins whose levels are significantly altered by mechanical stress.

To follow up on these experiments, they use proximity labelling to identify proteins whose interaction with the UNC-45 chaperone is modified by stress. UNC-45 localizes to myofilaments and mediates myosin refolding upon muscle damage. Among the proteins they identify is the E3 ubiquitin ligase NHL-1, which is the focus of the second half of the paper. The authors show that NHL-1 interacts preferentially with misfolded muscle myosin, and provide evidence that in the absence of NHL-1, misfolded UNC-54 remains associated with sarcomeric components instead of being recycled by the unfolded protein response.

By studying knockdowns of *nhl-1*, as well as ligase-defective mutants, and using both biochemical and functional assays, the authors infer that NHL-1 E3 ligase activity affects myosin protein levels and functionality under conditions of myosin misfolded stress. Interestingly, disrupting NHL-1 E3 ligase activity both improves the locomotory phenotype of myosin misfolding mutants (presumably by reducing myosin degradation) and attenuates the vulval rupture and decreased tip-to-tip length of body wall muscle cells induced by optogenetic stimulation in otherwise wildtype animals (presumably because of reduced force generation in otherwise wild type muscle).

The manuscript is clearly written and will interest a broad audience. The authors effectively integrate analyses at different levels of abstraction – biochemical, cell biological and functional, to elucidate the role of NHL-1 in muscle repair following mechanical stress.

Main comments

1) The observation that many animals rupture at the vulva following optogenetic stimulation with OptIMMus introduces some potential confounds. It likely means that some of the changes they identify in their transcriptomic and proteomic experiments are not downstream of mechanical stress per se but reflect damage to the animal's integrity. The authors should discuss this caveat.

2) The authors fused UNC-45 to miniTurboID, but as a (cytoplasmic) control expressed a different biotin ligase, TurboID, in muscle cells. Why was this done? Different activities of these biotin ligases will likely create systematic differences in their data. The authors should discuss this.

In an ideal world, the authors would repeat the experiment in Fig 3, comparing UNC-45-miniTurboID data to cytoplasmic miniTurboID (instead of TurboID). I suspect this would not change their conclusions much, but the current scheme for controlling their is sub-optimal.

In mitigation of this comment, it is fair to say that the protein levels of UNC-45-miniTurboID and, in the more appropriate control experiment I advocate above, of cytoplasmic miniTurboID, may not be the same, due to different protein turnover rates. It is thus hard to

design a perfect control. In general, miniTurboID is reported to be a less active biotin ligase than TurboID, so the authors are more likely to be under-reporting than over-reporting.

Reporting on the performance of miniTurboID compared to TurboID would be useful for the *C. elegans* community.

3) As highlighted in (1) above, the proximity labelling also has the caveat that the change in proximity to UNC-45-miniTurboID may not reflect the effect of mechanical stress per se, but of vulval rupture as a result of mechanical stress. Ten 5s pulses caused rupture in 30 – 40% of animals; the 13 pulses used in the proximity labelling presumably cause higher rupture levels. Ruptured animals likely sit around for some time before being collected for analysis by proteomics. This should be discussed. Again, I do not think this caveat is likely to affect the conclusions of the main part of the paper, which focuses on NHL-1 function. To my mind it would have been preferable to reduce optogenetic stimulation to avoid rupture.

Minor comments

- 1) L170 and L172: Include adjusted p-value, not only p-value.
- 2) The bibliography has typographical errors e.g. species names are not italicized.
- 3) Sup. Fig 1: show speed before light pulse.
- 4) Sup. Fig S5: Can the authors explain why OptiMMuS treatment appears to reduce NHL-1 binding to wild type UNC-54?

Reviewer #2 (Remarks to the Author):

In this paper by Kutzner et al., the authors utilized a newly developed transgenic strain that allows for optogenetic activation of muscle cells, consistently inducing muscle contraction. They used the associated mechanical stress to investigate the molecular changes within muscles, with a specific focus on the myosin-related chaperone UNC-45. Through proximity labeling co-IP and MS analysis, they discovered a crucial role for a new TRIM E3 ligase, NHL-1, in maintaining muscle proteostasis.

The study elegantly demonstrates that reducing NHL-1 or using an NHL-1 mutant incapable of ubiquitination in a conditional *unc-54(ts)* mutant or muscles under mechanical stress rescues normal myosin levels and myosin function.

The study is a significant advancement in the field, as the channelrhodopsin-regulated induction of mechanical stress in muscle opens new possibilities to investigate muscle wasting associated with many diseases, yet at the same time the authors discover NHL-1 as a crucial regulator of muscle proteostasis that directly interacts with myosin.

However, there are some aspects that the study does not address. While there are extensive whole animal transcriptomic and proteomic analyses that identify differentially regulated transcripts and proteins when muscle is under mechanical stress, including the Ubiquitin proteasome system, it does not establish a relationship with the proteasome, which could further support their conclusion. While the enrichment score in Figure 5b suggests reduced proteasome ontology terms, and *unc-54* is clearly decreased in the *unc-54* conditional mutant, proteasome inhibition, could have a similar effect as *nhl-1* knockdown or use of ubiquitination-deficient *nhl-1* mutant (as shown in Figure 5e and 6b).

Furthermore, the continuous muscle contraction and the *unc-54* conditional mutant appear to be seemingly unrelated conditions. The study does not investigate whether myosin (*unc-54*)

levels are reduced upon optogenetic activation of muscle movement upon light pulses. To bring this aspect full circle, as is indicated in Figure 7, they demonstrate that the ubiquitylation-deficient *nhl-1* mutant rescues the ChR2-blue light-induced phenotype. However, it would be essential to validate these results showing that under these conditions levels of myosin are likewise reduced and that depleting the ubiquitin-proteasome system (UPS) confirms proteasomal degradation of myosin under these conditions.

A minor comment for text related to Figure 5: a bit more explanation about the deficiencies of the individual *unc-45* mutants, such as *m94* and *e286*, would be beneficial, as not all readers may be equally knowledgeable about the *unc-45* genetic alleles.

Reviewer #3 (Remarks to the Author):

The manuscript "Optogenetic induction of mechanical muscle stress identifies myosin regulatory ubiquitin ligase NHL-1" by Kutzner et al. established a novel model for mechanical stress using optogenetic induction of muscle contraction in *C. elegans*. Combining this model and an established model for myosin misfolding (*unc-54ts*), the author employed transcriptomic, proteomics, and proximity ligation under relaxed and stressed conditions to identify novel regulators of myosin proteostasis that interact with the myosin chaperone UNC-45.

The work is comprehensive, going from building and characterizing a new and highly regulated model for muscle-specific stress through omics tools to identify novel players to a mechanistic examination of the role of these proteins in the myosin proteostasis network. The new model, mutants, and tools significantly contribute to the field. Moreover, the identification and characterization of NHL-1 and F40A3.6 promote understanding of myosin quality control.

Minor comments:

- (1) Does OptIMMuS-triggered sustained contractions induce a muscle stress response? Namely, are HSP or other cytoprotective genes induced mainly in muscle? For example, do worms expressing an HS reporter, such as *hsp-16.2::GFP*, show induced GFP expression in muscle cells after the treatment?
- (2) It will be interesting to compare the proteomic impact of OptIMMuS activation to myosin misfolding (*unc54ts*). The manuscript "Global proteome metastability response in isogenic animals to missense mutations and polyglutamine expansions in aging" doi: <https://doi.org/10.1101/2022.09.28.509812> examined the impact of *unc54ts* using mass spec and can be used for this comparison.
- (3) In Fig 5. it is not clear at which temp experiments were performed. I assume 25C, given the motility rates, but it is not noted.
- (4) In Fig. 5e.f I assume the myosin levels are at 25C. How do the levels compare with 15C? Why is the increase (~2-fold) not close to WT levels (Fig. 6b)? Please address this in the discussion (lines 422-430).

Reviewer #4 (Remarks to the Author):

I am reviewing the manuscript "Optogenetic induction of mechanical muscle stress identifies myosin regulatory ubiquitin ligase NHL-1" by Hoppe and colleagues in consideration for

publication in Nature Communications. In their work, the authors expressed a step function opsin in body wall muscles cells which causes prolonged depolarization and contraction, from which they hypothesized adverse effects on muscle function. The authors performed a proximity ligation analysis using the well-known myosin chaperone unc-45 as a bait and uncover previously unrecognized functions of nhl-1 ubiquitin ligase in muscle function. I find the paper very well written, the methods well explained in great detail and the results quite interesting. The authors should be commended on their serious and thorough proteomic analyses. However, reading the manuscript was rather unwieldy as figures, figure legends and supplementary material was presented in separate section, which did not facilitate reading on the screen.

Major:

1) I have my reservation regarding the novelty of the presentation. In particular, the optogenetically induced contraction as a means to identify muscle defects under mechanical stress is not new and has been used in several studies before. I wonder why these results are not introduced in the paper, e.g PMID: 26822332, who designed a microfluidic chip aiding quantification of body length change. That being said, one might conclude that the only new things on this method is the name OptIMMuS.

2) I do see the logic of inferring mechanical stresses from changes in body length after induction of muscle contraction. However, in that particular case, am reluctant to directly relate the effect on mitochondria and possibly other phenotypes on mechanical stress, as the optogenetic activation also induces long lasting depolarization of the excitable muscles and concomitant Ca signaling and also may generate significant phototoxicity over repetitive stimulation at 455nm. Along those lines, a similar channelrhodopsin with a prolonged open state lifetime causes neuronal degeneration (PMID: 37024651). Thus, an important control would be to show that the presented phenotypes are indeed due to mechanical stress and NOT due to Calcium overload. This could be done either by directly measuring/visualizing stress on mitochondria, and/or repeating the quantification in an unc-54 mutant that cannot contract. Do the authors find the same UNC-45 targets upon acute, but passive mechanical stress?

Minor:

1) In all figures, p-values should be exact, wherever reasonable, and not indicated by asterisks.

2) The power for the optogenetic stimulation should be given exact and for each experiment. Greater than is not a sufficient description of the experimental parameters.

3) I was not able to access the proteomic dataset under the links given.

Reviewers' Comments and our *Point-by-Point Response*

Reviewer #1:

The manuscript is clearly written and will interest a broad audience. The authors effectively integrate analyses at different levels of abstraction – biochemical, cell biological and functional, to elucidate the role of NHL-1 in muscle repair following mechanical stress.

Thanks to the reviewer and we are grateful for the positive feedback on our work.

Main comments

Point 1: The observation that many animals rupture at the vulva following optogenetic stimulation with OptIMMuS introduces some potential confounds. It likely means that some of the changes they identify in their transcriptomic and proteomic experiments are not downstream of mechanical stress per se but reflect damage to the animal's integrity. The authors should discuss this caveat.

We thank the reviewer for raising this relevant issue. In the Methods section, lines 898-900, we explain the sample collection procedure, which removes severely damaged and dead animals by sedimentation: "During sample collection after OptIMMuS, broken or dead worms were separated by washing the worms in M9 buffer, allowing them to settle, and discarding the supernatant with floating dead worms, leaving only intact worms in the pellet and in subsequent analyses." This procedure was performed prior to any downstream analyses such as fixation for microscopy, Western blot, transcriptomics, and proteomics. We agree that this information should appear earlier in the manuscript text and have added a statement in the Results section, line 110 (now line 104): "During sample collection and before any downstream analyses, these ruptured animals were removed by sedimentation in M9 buffer, as they did not settle and could be discarded with the supernatant."

Point 2: The authors fused UNC-45 to miniTurboID, but as a (cytoplasmic) control expressed a different biotin ligase, TurboID, in muscle cells. Why was this done? Different activities of these biotin ligases will likely create systematic differences in their data. The authors should discuss this. In an ideal world, the authors would repeat the experiment in Fig 3, comparing UNC-45-miniTurboID data to cytoplasmic miniTurboID (instead of TurboID). I suspect this would not change their conclusions much, but the current scheme for controlling their is sub-optimal. In mitigation of this comment, it

is fair to say that the protein levels of UNC-45-miniTurboID and, in the more appropriate control experiment I advocate above, of cytoplasmic miniTurboID, may not be the same, due to different protein turnover rates. It is thus hard to design a perfect control. In general, miniTurboID is reported to be a less active biotin ligase than TurboID, so the authors are more likely to be under-reporting than over-reporting. Reporting on the performance of miniTurboID compared to TurboID would be useful for the *C. elegans* community. *We agree with the reviewer that this is an important issue to address. We used transgenic worm strains expressing UNC-45-miniTurbo-HA and TurboID-HA and similar levels of the ligase protein, which resulted in comparable biotinylation over background (see additional panels d and e in Supplementary Fig. S4). Branon et al. (2018, doi:10.1038/nbt.4201) has reported that the activity of miniTurbo is lower than that of TurboID in adult worms. The authors further mention that "miniTurbo is less stable than TurboID (likely due to removal of its N-terminal domain)," which we can confirm, as we were unable to isolate a strain expressing miniTurbo-HA alone without an N-terminal tag protecting its truncated N-terminus from turnover. This limitation was circumvented by fusing UNC-45 N-terminally to miniTurbo-HA, thereby increasing its stability. We have now added a statement on the performance of miniTurbo and TurboID in line 209 (now line 205): "Both the fusion biotin ligase UNC-45-miniTurbo and the free biotin ligase TurboID were expressed at similar levels and produced a comparable biotinylation pattern that was enhanced over endogenous background biotinylation (Supplementary Fig. S4d, e)".*

Point 3: As highlighted in (1) above, the proximity labelling also has the caveat that the change in proximity to UNC-45-miniTurboID may not reflect the effect of mechanical stress per se, but of vulval rupture as a result of mechanical stress. Ten 5s pulses caused rupture in 30 – 40% of animals; the 13 pulses used in the proximity labelling presumably cause higher rupture levels. Ruptured animals likely sit around for some time before being collected for analysis by proteomics. This should be discussed. Again, I do not think this caveat is likely to affect the conclusions of the main part of the paper, which focuses on NHL-1 function. To my mind it would have been preferable to reduce optogenetic stimulation to avoid rupture. *We agree with the reviewer that vulval rupture and the inclusion of damaged or dead animals in our analyses could be a caveat of the experimental design. As described in the response to Point 1, a cleaning step had been included in the sample collection procedure to mitigate this issue by removing floating dead animals during washing steps prior to downstream analyses. We have now highlighted this fact earlier in the text*

in line 110 (now line 104): "During sample collection and before any downstream analyses, these ruptured animals were removed by sedimentation in M9 buffer, as they did not settle and could be discarded with the supernatant."

Minor comments

Point 1: L170 and L172: Include adjusted p-value, not only p-value.

The statement in lines 170-172 describes the filtering strategy after transcriptomics and proteomics data integration and before GO analysis. To include enough hits in the lists for an effective GO term overrepresentation analysis, it was necessary to filter less strictly and to use the p-value instead of the adjusted p-value.

Point 2: The bibliography has typographical errors e.g. species names are not italicized.

We have corrected typos and italics in the references.

Point 3: Sup. Fig 1: show speed before light pulse.

We did not measure the worm velocity before the first 5 s light pulse. The first data point in Supplementary Fig. S1a represents the mean worm velocity during the first 2 min of the measurement, which includes the 5-s blue light pulse and 115 s of red light illumination to simulate darkness. That the worms are motile prior to a blue light pulse can be seen in the first 5 s of Supplementary Video V1 prior to a 5-s blue light pulse (indicated by the video image becoming brighter), which stops the movement of the worms, contracts and paralyzes them, and results in the expulsion of eggs from their vulva.

Point 4: Sup. Fig S5: Can the authors explain why OptiMMuS treatment appears to reduce NHL-1 binding to wild type UNC-54?

We thank the reviewer for raising this point. Some observed differences in binding efficiency in the UNC-54/MHCB/myosin and UNC-45 co-immunoprecipitation experiments may be due to the transient nature of NHL-1 binding, which has not been detected by other conventional interaction screens. Slight pipetting differences and altered protein input amounts seem to affect the efficiency of NHL-1 binding in the immunoprecipitation experiments when compared to the input lanes in supplemental Fig. S5b and c. In support of this notion, only proximity labeling of transient interactors under mechanical stress and mass spectrometry was initially able to identify NHL-1, which was missed in previous studies. However, complementary co-immunoprecipitation experiments using monoclonal or polyclonal antibodies against either FLAG-NHL-1, UNC-54/myosin, or UNC-45 clearly revealed the protein

complex UNC-54/MHCB/myosin - UNC-45 - (FLAG-)NHL-1 - (HA-)F40A3.6 in slightly different compositions and concentrations (Fig. 4c, Supplementary Fig. S5b, c), which is even increased in strains expressing the unc-54(e1301) myosin mutation.

Reviewer #2:

Point 1: The study elegantly demonstrates that reducing NHL-1 or using an NHL-1 mutant incapable of ubiquitination in a conditional unc-54(ts) mutant or muscles under mechanical stress rescues normal myosin levels and myosin function. The study is a significant advancement in the field, as the channelrhodopsin-regulated induction of mechanical stress in muscle opens new possibilities to investigate muscle wasting associated with many diseases, yet at the same time the authors discover NHL-1 as a crucial regulator of muscle proteostasis that directly interacts with myosin. However, there are some aspects that the study does not address. While there are extensive whole animal transcriptomic and proteomic analyses that identify differentially regulated transcripts and proteins when muscle is under mechanical stress, including the Ubiquitin proteasome system, it does not establish a relationship with the proteasome, which could further support their conclusion. While the enrichment score in Figure 5b suggests reduced proteasome ontology terms, and unc-54 is clearly decreased in the unc-54 conditional mutant, proteasome inhibition, could have a similar effect as nhl-1 knockdown or use of ubiquitination-deficient nhl-1 mutant (as shown in Figure 5e and 6b).

We appreciate the positive feedback on our work. In previous publications, which are also cited in the Introduction (lines 64-70, ref21: Landsverk et al. 2007, doi:10.1083/jcb.200607084 and ref23: Lehrbach and Ruvkun 2019, doi:10.7554/eLife.44425), it was established that misfolded myosin is degraded by the proteasome: UNC-54/MHCB/myosin degradation in worm lysate is promoted by induction of proteasome activity by addition of ATP and attenuated by proteasome inhibition by addition of MG132 (see Fig. 3B in ref21); furthermore, genetically induced myosin misfolding in the unc-54(e1301) strain and other myosin misfolding strains activates the expression of the proteasomal subunit reporter rpt-3p::GFP via SKN-1A (see fig. 1 in ref23). Indeed, we also observed a stabilization of UNC-54 protein levels in the unc-54(e1301) myosin misfolding mutant upon RNAi against proteasomal subunit genes such as rpn-8 and rpn-11 (see Figure I).

Figure I Proteasomal inhibition by RNAi against proteasomal subunits starting from L4 stage increases myosin protein levels in the myosin misfolding mutant *unc-54(e1301)* at the restrictive temperature of 25°C.

Point 2: Furthermore, the continuous muscle contraction and the *unc-54* conditional mutant appear to be seemingly unrelated conditions. The study does not investigate whether myosin (*unc-54*) levels are reduced upon optogenetic activation of muscle movement upon light pulses. To bring this aspect full circle, as is indicated in Figure 7, they demonstrate that the ubiquitylation-deficient *nhl-1* mutant rescues the Chr2-blue light-induced phenotype. However, it would be essential to validate these results showing that under these conditions levels of myosin are likewise reduced and that depleting the ubiquitin-proteasome system (UPS) confirms proteasomal degradation of myosin under these conditions.

*We thank the reviewer for raising this important point. Our OptIMMuS intervention in C. elegans is designed to recapitulate intense, unaccustomed exercise in humans. It has been previously reported that both intense exercise conditions and muscle atrophy result in varying degrees of myofiber damage and increased expression of ubiquitin, proteasomal subunits, and myosin-targeting E3 ligases (Murton et al. 2008, doi:10.1016/j.bbadis.2008.10.011). Thus, UPS activation appears to be required for muscle adaptation and myofilament remodeling in both conditions. Regarding myosin levels in OptIMMuS, we observed that myosin (MHC B/UNC-54) was not significantly changed at the protein level (log2 fold change = 0.04, adjusted p-value = 0.893723), but slightly increased at the transcript level (log2 fold change = 0.22, adjusted p-value = 0.001485) in whole worm transcriptomics and proteomics. This observation could indicate a continuous turnover of individual damaged myosin molecules and recovery of protein levels by increased transcription. Accordingly, on Western blots of UNC-54/myosin after 10 5-second pulses of blue light in OptIMMuS in the wild-type and *nhl-1(syb8175)* RING domain mutant background with and without bortezomib treatment, we were unable to detect obvious reproducible differences in myosin protein levels (see Figure II), although differences in muscle contractility could be observed in OptIMMuS (Fig. 6 e-g and h-*

j). We have included new data on vulval rupture and body wall muscle cell length shortening in OptIMMuS after proteasome inhibition with bortezomib in the Results section, lines 345-355 and Fig. 6 h-j. A likely explanation, which informed the design of the model in Figure 7, is that during sustained contractions, only individual myosin molecules unfold and misfold locally in the sarcomere, requiring refolding by UNC-45 and/or targeting for degradation by NHL-1, which we are unable to detect by Western blot. In myosin misfolding genetic mutants, however, myosin unfolding occurs on a more global scale, making the effect significant and detectable by Western blot. As explained in lines 381-389 of the Discussion section, the combined data from OptIMMuS and genetic mutant experiments together support the idea that myosin misfolds in the sarcomere, is ubiquitinated by NHL-1, and is targeted for proteasomal degradation.

Figure II Myosin protein levels are unchanged after 10 5-s blue light pulses in wild-type and *nhl-1(syb8175)* RING domain mutant background (a, b) in two independent experiments, (c) even when proteasomal degradation is inhibited by the dipeptide drug bortezomib (5 μ M).

Point 3: A minor comment for text related to Figure 5: a bit more explanation about the deficiencies of the individual unc-45 mutants, such as m94 and e286, would be beneficial, as not all readers may be equally knowledgeable about the unc-45 genetic alleles.

We have updated the Results section in line 317 to include the following statement: "To date, no apparent difference in the phenotypes and mechanisms of the two close UCS domain mutants m94 and e286 has

been reported. Both ts mutations result in myosin misfolding and reduction at the protein level with a compensatory increase in UNC-45 protein levels at the restrictive temperature of 25°C (Moncrief et al. 2021). In crystal structures, the UNC-45(e286) protein has been reported to have a more rigid myosin-binding UCS domain that is moved closer to the central domain (Hellerschmied et al. 2019). In the unc-45(b131) background, where UNC-45 binding to NHL-1 is normal but binding to UNC-54/myosin is reduced due to structural remodeling of the myosin-binding canyon (Hellerschmied et al. 2019; Supplementary Fig. S6c, d), depletion of nhl-1 and F40A3.6 had no significant effect (Fig. 5c)."

Reviewer #3:

The work is comprehensive, going from building and characterizing a new and highly regulated model for muscle-specific stress through omics tools to identify novel players to a mechanistic examination of the role of these proteins in the myosin proteostasis network. The new model, mutants, and tools significantly contribute to the field. Moreover, the identification and characterization of NHL-1 and F40A3.6 promote understanding of myosin quality control.

We would like to thank the reviewer for the kind assessment of our work.

Minor comments

Point 1: Does OptIMMuS-triggered sustained contractions induce a muscle stress response? Namely, are HSP or other cytoprotective genes induced mainly in muscle? For example, do worms expressing an HS reporter, such as hsp-16.2::GFP, show induced GFP expression in muscle cells after the treatment?

We thank the reviewer for this interesting question. In whole worm transcriptomics, we observed no significant change in the hsp-16.2 gene, and of the other hsp-16 family genes, only hsp-16.11 was significantly altered (log2 fold change = -0.827696112605779, adjusted p-value = 0.000352515749778064). In whole worm proteomics, the two detected HSP-16 sHSPs were slightly but not significantly increased: HSP-16.48 (log2 fold change = 0.848073, adjusted p-value = 0.385855), HSP-16.11 (log2 fold change = 0.123577, adjusted p-value = 0.818162).

Point 2: It will be interesting to compare the proteomic impact of OptIMMuS activation to myosin misfolding (unc54ts). The manuscript "Global proteome metastability response in isogenic animals to missense mutations and polyglutamine expansions in aging" doi: <https://doi.org/10.1101/2022.09.28.509812> examined the

impact of unc54ts using mass spec and can be used for this comparison.

We thank the reviewer for raising this interesting point. Consistent with our proximity labeling, the mentioned manuscript (ref44, Sui et al. 2022, doi:10.1101/2022.09.28.509812) also found that the HSP-16 family sHSPs and UNC-45 were increased on total protein and decreased in PK accessibility in unc-54(e1301) mutant worms, whereas UNC-54/myosin itself was increased in PK accessibility. The latter finding suggests myosin unfolding, whereas the protein increase and concomitant decrease in PK accessibility of chaperone proteins suggests their oligomerization and binding. We have updated the Results section in line 220 (now line 218-221) to read as follows: "Previously, transcriptional induction of these hsp-16 heat shock response genes had been observed in unc-45 knockdown (ref40, Schmauder & Richter. 2021) and an increase in HSP-16 family proteins with concomitant conformational rigidity had been observed in the myosin misfolding mutant unc-54(e1301) already at the permissive temperature (ref44, Sui et al. 2022)".

Point 3: In Fig 5. it is not clear at which temp experiments were performed. I assume 25C, given the motility rates, but it is not noted. In the Results section, lines 311 say "at the restrictive temperature", which has now been clarified to say, "at the restrictive temperature of 25°C".

Point 4: In Fig. 5e.f I assume the myosin levels are at 25C. How do the levels compare with 15C? Why is the increase (~2-fold) not close to WT levels (Fig. 6b)? Please address this in the discussion (lines 422-430).

As described in the manuscript mentioned above (ref44, Sui et al. 2022, doi:10.1101/2022.09.28.509812), unc-54(e1301) myosin misfolding worms appear to be only pseudo-wild-type at the permissive temperature of 15°C. Accordingly, already at 15°C, myosin protein levels are slightly decreased compared to the wild type and UNC-45 protein levels are increased, probably as a compensatory response (see Figure III). This effect is even more pronounced at the restrictive temperature of 25°C, when myosin misfolding is enhanced and the corresponding worms exhibit myofibrillar disorganization and a motility defect and has also been described for unc-45 temperature-sensitive mutants that exhibit myosin misfolding (ref18, Moncrief et al. 2021, doi:10.1002/pro.4180).

Figure III *unc-54(e1301)* myosin misfolding mutants are pseudo-wild-type. Already at the permissive temperature of 15°C, UNC-54/myosin protein levels are slightly reduced compared to wild-type.

As discussed in lines 422-430, the reason why nhl-1 knockdown or mutation is unable to restore myosin levels in myosin misfolding mutants to wild-type or permissive temperature levels is likely due to the engagement of additional E3 ligases and different degradation pathways to compensate for the loss of one E3 ligase, as is the case in the mammalian system for the three MuRF E3 ligases, Atrogin-1 and others. To clarify and explain, we have updated the Discussion section in lines 422-430 (now line 433-442): "In contrast, knockdown or RING domain mutation of NHL-1 partially rescues both motility and myosin/UNC-54 protein levels in both unc-45(m94) and unc-54(e1301) mutants (Figures 5c, e, f and 6b, c, d), suggesting that myosin misfolding is the underlying trigger for NHL-1 to act on myosin protein levels and functionality. We propose that UFD-2 specifically regulates the myosin-directed chaperone UNC-45, whereas NHL-1 directly regulates myosin during mechanical stress, myosin misfolding, and muscle atrophy. Nevertheless, considering the diverse and redundant number of mammalian E3 ligases that regulate myosin, it is likely that there are different ubiquitylation pathways that target myosin for degradation under different physiological conditions, which may also explain why the increase in myosin protein levels by nhl-1 knockdown or mutation does not fully restore wild-type levels."

Reviewer #4:

I find the paper very well written, the methods well explained in great detail and the results quite interesting. The authors should be commended on their serious and thorough proteomic analyses. However, reading the manuscript was rather unwieldy as figures, figure legends and supplementary material was presented in separate section, which did not facilitate reading on the screen.

We thank the reviewer for the concise summary of the results and the praise of the methodological execution. We sincerely apologize for any inconvenience caused by the order of our figures and figure legends in this first manuscript submission.

Main comments

Point 1: I have my reservation regarding the novelty of the presentation. In particular, the optogenetically induced contraction as a means to identify muscle defects under mechanical stress is not new and has been used in several studies before. I wonder why these results are not introduced in the paper, e.g PMID: 26822332, who designed a microfluidic chip aiding quantification of body length change. That being said, one might conclude that the only new things on this method is the name OptIMMuS.

We thank the reviewer for bringing this to our attention. The referenced study (Hwang et al. 2016, doi: 10.1038/srep19900) is a muscle contraction phenotypic analysis that used optogenetics to describe the kinetics of a single muscle contraction and relaxation event in sarcomere component mutants with only mild or no defects in sarcomere structure. The optogenetic strain used in this study expressed the ChR2 gain-of-function variant H134R in cholinergic motor neurons and had previously been characterized in Liewald et al. 2008 (doi:10.1038/nmeth.1252) for use in the study of neurotransmission. In both studies, optogenetics was used for experiments consisting of a single or a few short contractions followed by microscopy or electrophysiology of individual worms. In contrast, the ChR2 variant C128S;H134R used in our OptIMMuS model is a double mutant channel with slowed closing kinetics that is stably expressed directly in the body wall muscle. This particular variant had been extensively characterized as a body wall muscle mosaic transgene expressed from an extrachromosomal array in the "Rhodopsin optogenetic toolbox v2.0" study by the Gottschalk lab (ref33, Bergs et al. 2018, doi:10.1371/journal.pone.0191802). Our stably integrated expression of this ChR2 variant in each body wall muscle cell allows the induction of repeated, intense, sustained contractions in large synchronized populations of isogenic worms over the course of several hours, thereby avoiding neuronal desensitization. Unlike previous studies using different ChR2 variants and neuronal expression, our OptIMMuS model allows the analysis of long-term biological effects of these intense, repetitive, and sustained contractions in synchronized populations rather than single individuals, using not only microscopy and electrophysiology, but also large-scale behavioral, omics, and screening approaches. This expanded range of applications warrants a shortened name to facilitate adaptation of this model in other labs for future studies. To compare our model with previous studies, we have now included the mentioned study in the Discussion section, lines 362-365 (now line 370-376): "A previous study used light-induced contraction by a fast neuronal ChR2 variant to describe the effect of sarcomere component mutants on the mechanics

and kinetics of a single muscle contraction and relaxation event (added as ref47, Hwang et al. 2016, doi:10.1038/srep19900). In contrast, our data demonstrate that the OptIMMuS model can be used to analyse not only cellular responses in single cells and tissues but also organismal physiology and phenotypes of synchronized worm populations. C. elegans genetic mutants crossed into the OptIMMuS strain define sensitized backgrounds that allow the study of response mechanisms to repeated intense muscle contractions otherwise missed under conventional experimental conditions. [...]"

Point 2: I do see the logic of inferring mechanical stresses from changes in body length after induction of muscle contraction. However, in that particular case, am reluctant to directly relate the effect on mitochondria and possibly other phenotypes on mechanical stress, as the optogenetic activation also induces long lasting depolarization of the excitable muscles and concomitant Ca signaling and also may generate significant phototoxicity over repetitive stimulation at 455nm. Along those lines, a similar channelrhodopsin with a prolonged open state lifetime causes neuronal degeneration (PMID: 37024651). Thus, an important control would be to show that the presented phenotypes are indeed due to mechanical stress and NOT due to Calcium overload. This could be done either by directly measuring/visualizing stress on mitochondria, and/or repeating the quantification in an unc-54 mutant that cannot contract. Do the authors find the same UNC-45 targets upon acute, but passive mechanical stress?

In our initial analysis of the OptIMMuS model, we did indeed focus on phenotypes directly related to the mechanical stimulus of repeated sustained contractions (Fig. 1). Since muscle contraction induces several types of stress, such as mechanical, thermal, and oxidative stress, we next examined possible evidence of general cellular and organismal stress resulting from our treatment (Fig. 2). Here we observed mitochondrial fragmentation, increased occurrence of autophagosomes, and fatigue, which we attribute to our OptIMMuS treatment. As the reviewer correctly points out, and as reported by Momma et al. (2017, doi:10.1534/genetics.117.202747), calcium overload in muscle due to (heat) stress can also lead to mitochondrial fragmentation, motility defects, and premature death. In general, mitochondrial fragmentation has been described as a response to various types of cellular stress (ref37, Youle et al. 2012), so we must rightly modify the corresponding statement in the Results section lines 148-150 to assign this observation to a general response to stress (now lines 144-146): "Taken together, these data demonstrate cellular stress phenotypes in body wall muscle

cells in response to optogenetically triggered repeated sustained contractions". Because calcium, energy production, and contraction are inextricably linked in the excitation-contraction coupling in muscle, muscle contraction cannot be separated from calcium homeostasis and possible secondary effects on other organelles when studying the stress of active contraction by myosin activity. To test whether the muscle cells are still viable and calcium homeostasis is still intact after 10 5-s pulses of blue light (the 3-h OptIMMuS treatment we use in most experiments), we verified that the worms could still move and contract their body wall muscles after treatment. As seen in Supplementary Video V2, we observed similar body contraction, movement paralysis, and egg expulsion in the uninjured worms for the 11th 5-s blue light pulse, indicating viable muscle structures, intact calcium homeostasis, and retained ChR2 stimulation capacity (Supplementary Video V2). In addition, as described in the response to reviewer 1's comment (1), we had included a cleaning step in the sample collection procedure to account for the severely damaged and possibly dead animals in our experiments by removing the ruptured animals that float in the supernatant and resist sedimentation during washing steps prior to downstream analyses. We have now highlighted this fact earlier in the text in line 110 (now line 104): "During sample collection and before any downstream analyses, these ruptured animals were removed by sedimentation in M9 buffer as they did not settle and could be discarded with the supernatant." To avoid the cellular degeneration observed for the neuronally expressed ChR2 variant L132C;H134R;T159C (see Supplementary Fig. 10 in the aforementioned study by Porta-de-la-Riva et al. 2023, doi:10.1038/s41592-023-01836-9), we also expressed the used ChR2 variant C128S;H134R directly in body wall muscle cells from the myo-3 promoter. In line, in fluorescence and electron microscopy, we observed that the muscle cells were intact, although their shape was altered (see Fig. 1g and Fig. 2a-c). We can also rule out phototoxic effects because we use worms grown without supplementation of the cofactor all-trans retinal (ATR) and exposed to blue light in parallel as a control in all our experiments. Next, to identify more specific stress responses to mechanical stress induced by OptIMMuS, we performed transcriptomics and proteomics experiments on synchronized populations of OptIMMuS-treated worms. Strikingly, muscle and sarcomeric proteins were specifically upregulated, in contrast to a general downregulation of general translation and biosynthesis processes, which may indicate a muscle-specific proteostasis response (Fig. 3). To further analyse the effects of mechanical stress induced by active contractions on the myofilament landscape and the motor protein myosin directly, we turned to proximity proteomics of the myosin chaperone UNC-45 (Fig. 4). We

investigated the role of the identified interaction of NHL-1 with UNC-45 and myosin using multiple approaches, including the orthogonal Y2H system and genetic mutants without OptIMMuS-induced perturbations (Fig. 5 and 6). As explained in the Discussion section lines 381-389, only the combined data of genetic mutants and proximity labeling can adequately verify that our OptIMMuS model indeed induces mechanical stress that leads to myosin misfolding in the muscle and a muscle-specific proteostasis response. Since calcium signaling is required for muscle contraction, its secondary effects cannot be easily uncoupled and are likely part of the complex response to contraction stress.

Minor comments

Point 1: In all figures, p-values should be exact, wherever reasonable, and not indicated by asterisks.

We have added the exact p-values to the figures.

Point 2: The power for the optogenetic stimulation should be given exact and for each experiment. Greater than is not a sufficient description of the experimental parameters.

The OptIMMuS incubator is equipped with two 100 W LED arrays (455 nm) spaced 24 cm apart and 30 cm above the sample tray (size: 36 cm x 54 cm) on which the upturned worm plates are randomly placed for each experiment. Light intensity on the sample tray was measured with a ThorLabs PM100D power and energy meter using a S120C photodiode power sensor (400-1100 nm, 50 mW) at a wavelength setting of 450 nm (range 1.9 mW) during incubator setup and several times during subsequent years of use. For reference, the light intensity during a 5-s pulse of blue light was measured to be consistently in the range of 10-16 $\mu\text{W per mm}^2$ at various positions on the sample tray. According to the extensive characterization of ChR2 variants in ref33, the contraction intensity induced by the channelrhodopsin variant ChR2(C128S;H134R) does not significantly increase above a light intensity of 5 $\mu\text{W per mm}^2$ (see Fig. 2e in Bergs et al. 2018, doi:10.1371/journal.pone.0191802). Therefore, we used the described setup for all our experiments. For a more detailed explanation of our setup, we have included the relevant information in the Methods section, lines 889-895. For supplemental microscopy videos, the 5-s blue light pulse was manually generated with a CoolLED pE-300ultra light source using the blue LED (450 nm) and a liquid light guide. The light intensity was adjusted to the light intensity of 10 $\mu\text{W per mm}^2$ inside the OptIMMuS incubator by parallel measurement with the ThorLabs Power and Energy Meter PM100D with a S120C photodiode power sensor (400-1100 nm, 50 mW) at the wavelength setting of 450 nm (range 1.9 mW). We have added this information to the

Supplementary Methods section in line 141 (now line 150) of the Supplement.

Point 3: I was not able to access the proteomic dataset under the links given.

The proteomic raw data files are accessible by logging in to the PRIDE database on the website (<https://www.ebi.ac.uk/pride/login>) using username and password provided as "Reviewer account details". Relevant transcripts and proteins identified in our analyses are also presented with statistical data in Supplementary Tables 1, 2, 4a, 4b, and 5.

REVIEWERS' COMMENTS

Reviewer #1 (Remarks to the Author):

The reviewers have addressed my comments. It may not hurt to add a sentence somewhere saying that 'We cannot exclude that some of the proteomic changes we detect reflect not only mechanical stress, but also the prolonged Ca²⁺ influx associated with OptIMMus.'

Reviewer #2 (Remarks to the Author):

In this revised version and the additional information in the rebuttal letter, the authors have clarified my question about the proteasome. In fact they have addressed all comments of all reviewers and this has further improved already excellent work and the manuscript.

Reviewer #3 (Remarks to the Author):

The Authors addressed all my comments and questions, as well as those of the other reviewers.

Reviewer #4 (Remarks to the Author):

The authors responded to all my questions. I have no further concerns.